# CRISPR-powered quantitative keyword search engine in DNA data storage

Jiongyu Zhang[1,2], Chengyu Hou[1,2] & Changchun Liu ®[1] ✉

Despite the growing interest of archiving information in synthetic DNA to confront data explosion, quantitatively querying the data stored in DNA is still a challenge. Herein, we present Search Enabled by Enzymatic Keyword Recognition (SEEKER), which utilizes CRISPR-Cas12a to rapidly generate visible fluorescence when a DNA target corresponding to the keyword of interest is present. SEEKER achieves quantitative text searching since the growth rate of fluorescence intensity is proportional to keyword frequency. Compatible with SEEKER, we develop non-collision grouping coding, which reduces the size of dictionary and enables lossless compression without disrupting the original order of texts. Using four queries, we correctly identify keywords in 40 files with a background of ~8000 irrelevant terms. Parallel searching with SEEKER can be performed on a 3D-printed microfluidic chip. Overall, SEEKER provides a quantitative approach to conducting parallel searching over the complete content stored in DNA with simple implementation and rapid result generation.

As the world enters the digital era, it is predicted that the global demand for data storage will reach $1.75 \times 10^{14}$ GB by 2025 and continue to skyrocket in the next few decades[1]. Traditional electro-mechanical storage devices, such as hard drives, are facing challenges in recording such an unprecedented amount of data due to their limited storage density and capacity. The use of DNA has emerged as a potential digital storage medium to meet the era of information explosion[2–9]. DNA has been proven to be superior to conventional storage media in terms of physical density[5,8,10], data longevity[4,11,12], and even information encryption ability[13]. Studies have reported that with a novel encoding scheme, perfect retrieval of information can be achieved from a density of 215 petabytes per gram of DNA[8], more than 88 million times higher than that of commercially available one-terabyte hard drives. The extreme durability of DNA is evidenced by the recovery of fossils from over one million years ago[14,15]. By contrast, the lifespan of hard drives can be potentially influenced by environmental conditions and mechanical failure.

Polymerase chain reaction (PCR) enables oligos sharing the same primer set to be selectively amplified before sequencing, hence the access to data can be random and programmed[16]. However, random access is typically based on unique file identifications (IDs)

rather than the actual content, preventing access to files of interest without prior knowledge of the link between the content and IDs. Considering that DNA storage is more often applied to infrequently accessed data, people may find it redundant to classify the data in terms of the actual content, as these categories may occupy significant secondary memory. Therefore, it is of great importance to develop a simple yet effective method to allow content-based searching directly conducted in the storage medium, with a highly accurate and easily comprehensible return of results. Researchers have proposed several approaches to enable searching in DNA drives[17,18], but these methods are often hybridization-based, which may require careful selection of orthogonal sequences representing different queries to ensure specificity.

Clustered regularly interspaced short palindromic repeats (CRISPR) is an acquired immune mechanism discovered in prokaryotes[19,20] that can identify a specific infectious DNA sequence in a cell overwhelmed with interfering genes, analogous to keyword search in a database. This similarity inspired us to broaden the application of CRISPR from genome editing[21–24] and molecular diagnostics[25–31] to a search engine for data stored in DNA. We leveraged the trans-cleavage activity[26] of CRISPR-Cas12a, which can be triggered

[1]Department of Biomedical Engineering, University of Connecticut Health Center, Farmington, CT 06030, USA. [2]Department of Biomedical Engineering, University of Connecticut, Storrs, CT 06269, USA. ✉e-mail: chaliu@uchc.edu

immediately upon recognition of a short DNA target sequence complementary to a CRISPR RNA (crRNA) query and lead to massive indiscriminate degradation of single-stranded DNA fluorophore-quencher (ssDNA-FQ) reporters, producing a visible search result. This method relies on enzymatic activity, which is less tolerable to nucleotide mismatches compared with hybridization-based approaches, and therefore may achieve higher specificity and fewer errors in keyword identification. We named this method Search Enabled by Enzymatic Keyword Recognition (SEEKER).

Herein, we used SEEKER to conduct keyword searches in abstracts of research papers in plain text format. The text data were encoded in DNA through the developed non-collision grouping (NCG) coding. We stored the dictionary in reference strands of the oligo pool with unique PCR primer IDs. The query sequence can be acquired only after the recovery of reference strands (typically 100–200 oligos) rather than the entire oligo pool. By integrating DNA data storage with microfluidics for easier operation, we also implemented SEEKER on three-dimensional (3D)-printed

microfluidic chips with lyophilized reagents. We found that the growth rate of fluorescence intensity was proportional to the frequency of the keyword in the file, enabling SEEKER to return quantitative search results, a function that has only been achieved with an electrical processor in the past.

## Results

### System overview

In general, a complete DNA data storage system with information searching capability should consist of three functional blocks (Fig. 1a): (1) Writing data into DNA. A major concern is allowing the mapping of characters to a fixed-length oligonucleotide query optimized for DNA hybridization and CRISPR-based detection. In this case, a dictionary becomes essential for storing the information regarding the relationship between text characters and nucleotide sequences. In our study, we compressed and stored the dictionary in ~100–200 reference strands through a pre-processing step to arrange the ASCII-coded data into groups without encountering collisions of "1"s in the same bit

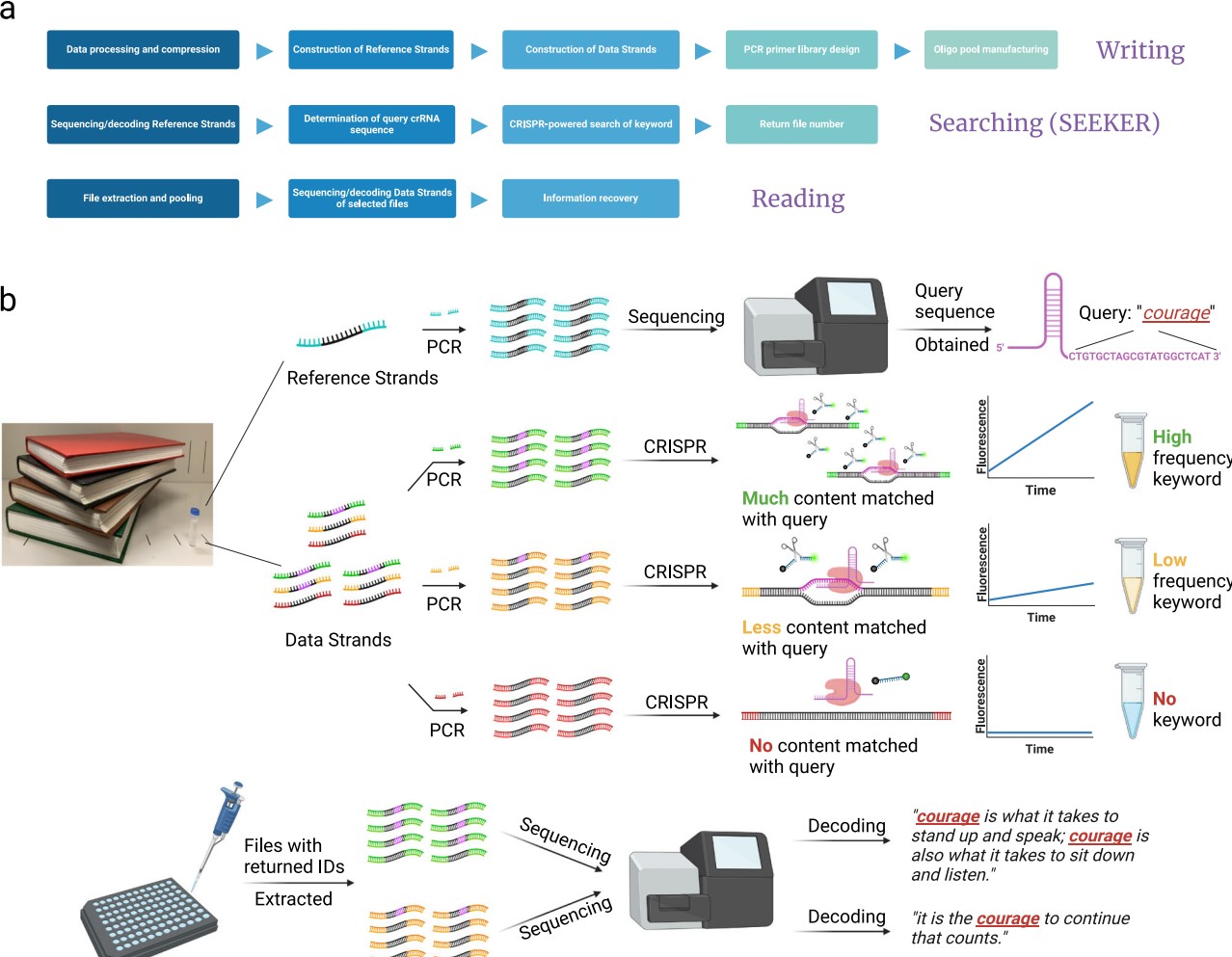

**Fig. 1 | Framework of a DNA data storage system with searching capability and the general workflow of SEEKER. a** Complete framework of a searchable DNA data storage system includes writing, searching, and reading the data. **b** The oligo pool storing text data is separately constructed into two parts: reference strands and data strands. The reference strands usually comprise 100–200 oligos and can be pre-sequenced to determine the dictionary used to map the data strands to binary codes as well as the crRNA spacer sequence of an intended query, for instance the keyword "courage" corresponds to the sequence "CTGTGCTAGCGTATGGCTCAT" in crRNA. The data strands are selectively amplified according to file IDs and then incubated with the Cas12a-crRNA ribonucleoprotein complex. The fluorescence

intensity increases rapidly if the amplified file contains many repeats of the keyword "courage", generating a strong fluorescence signal within a short time. If fewer instances of the keyword "courage" appear in the file, the fluorescence enhancement retards, and the endpoint fluorescence intensity becomes weaker. If the keyword "courage" is not found in the file, no fluorescence will be detected. After searching, files generating positive signals are recognized as carrying the data-of-interest and are subjected to next-generation sequencing to recover the complete content. In this example, a stronger signal is generated when the keyword "courage" appears twice, whereas a weaker signal is generated when the keyword only appears once. Illustrations were created with BioRender.com.

position. With the dictionary, characters in texts can be computationally mapped to nucleotides as a data unit, which are then concatenated to constitute a data strand. Both reference and data strands are sectioned into -150-bp pieces to meet the recommended read length for Illumina-based sequencing. Then, potential PCR primers screened for sensitivity and specificity are both prefixed and suffixed to each oligo as IDs for files or dictionaries (more than one dictionary can be present in a single oligo pool), and the sequences are submitted to manufacturing. (2) Searching for data-of-interest, which is accomplished through SEEKER as demonstrated in the following paragraph. (3) Reading data-of-interest through next-generation sequencing, where files returning positive results are pooled and sequenced. With complete recovery of the dictionary, the nucleotide sequences in the data strand are reversibly mapped to ASCII codes and then readable texts.

While encoding and decoding have been well studied, implementing search functions remains largely unexplored. In this study, we constructed SEEKER as a prototypical search engine for DNA data storage. SEEKER conducts search on the entire text information stored in an oligo pool and returns visually interpretable results in a timely manner. Searching with SEEKER (Fig. 1b) typically requires a computational determination of query sequences, which includes PCR amplification and sequencing of reference strands and algorithmic mapping of the ASCII code of a query to a nucleotide sequence. Then, the files are amplified and the Cas12a system is introduced where crRNA works as the query. If keywords are abundant, complementary base pairing between the query and content triggers strong collateral activity of Cas12a and results in massive cleavage of ssDNA-FQ, generating distinguishable fluorescence within a short period. The enhancement of fluorescence decelerates as the keyword becomes less frequent in amplicons. At the endpoint of detection, the fluorescence intensity for a rarely appearing keyword is milder than that of an abundant keyword. If no keyword is found in the content, the collateral activity is not initiated, and no fluorescence enhancement can be observed as the fluorophores remain quenched.

Long queries, such as phrases and sentences, are also valid for SEEKER. This can be achieved by dividing the long query into several short queries and building a reaction array on grid-patterned microchambers with pre-stored CRISPR reactions corresponding to each individual query (Supplementary Fig. 1a). The SEEKER system is also designed capable of detecting word permutations in a long query by searching with queries encoding the "junctions" of words (Supplementary Fig. 1b). The growth rate of fluorescence is proportional to the frequency of the keywords, rendering SEEKER a quantitative method for lexical searching of information stored in DNA. Lastly, all target files are assembled in a new pool for next-generation sequencing, after which the reads are converted to binary data and further translated to intelligible texts.

This search scheme is similar to a linear search, which is the simplest approach requiring every document to be opened and examined in order to search for the term (Supplementary Table 1). We employed this search scheme with the aim of selectively accessing many files to demonstrate the feasibility of SEEKER, primarily because it allows us to validate the following: (1) files that are not supposed be accessed should not be amplified, testing the effectiveness of our orthogonal PCR primer set design; (2) content from unamplified files should not generate signals in the CRISPR reaction, preventing catastrophic misidentification; (3) the system should be reliable when using the same query to conduct searches in different files, ensuring stability in different noisy backgrounds. Searches based on metadata and data structures such as inverted indexing can dramatically improve efficiency. We further demonstrated that SEEKER can be adapted to such search schemes with slight modification (Supplementary Fig. 2).

## Oligo data structure and general encoding/decoding procedure

The establishment of reference strands and data strands follows the NCG coding algorithm (Supplementary Figs. 3–4). Given that sequencing of the dictionary is inevitable for determining the query sequence used for searching, the NCG coding provides a solution to shorten the codeword for the dictionary by "grouping" data fragments with no collisions of "1"s under the same index. We validated that, compared with a classical coding technique that solely relies on repetition elimination without the NCG mechanism, the number of nucleobases required for the dictionary can be reduced by up to -15% (Supplementary Fig. 4b). Moreover, as only a small portion of the files is meant to carry the keyword and should be sequenced, we demonstrated that the overall cost of writing-searching-reading process can be lower with data encoded in NCG (Supplementary Fig. 4d, Supplementary Table 2). A detailed analysis and discussion of this coding technique can be found in Section 2 of the Supplementary Information.

The structures of the reference strands and data strands are illustrated in Fig. 2a. The quantity of the reference strands is equal to the grouping interval, and the numbering of the reference strands corresponds to the position of the bit in a data fragment. The data strand is composed of multiple data units containing several bases, of which the last base is the pointer and the remaining bases correspond to the group index. For reference strands, two index units are prepended to indicate the reference number and the segment number, while only one is required for the data strand since the file ID has been encoded in the sequences of the PCR primer target. Based on the error rate in Illumina-based sequencing, which is reported to be around 1%[32,33], we appended two of the 6-nt Reed–Solomon (RS) error correction bases to each reference strand and data strand to detect up to 2 erroneous bases and correct up to 1 erroneous base in a sequencing read.

Following the primer screening criteria provided by Organick et al.[16]. and considering additional restrictions to avoid potential interferences on CRISPR-based identifications (Supplementary Fig. 5), we generated 185–205 orthogonal sequences as primer candidates in 100 trials of primer screening (Supplementary Fig. 6). We included different sets of 21-nt PCR primer target sequences at the beginning and the end of each data strand to serve as IDs of different files. The reference strands were also supplemented with a set of primer target sequences, which can be altered to represent different reference pools. Although a traditional primer selection strategy would only allow -100 files encoded with this scale of primer availability, we anticipate -10,000 files can be encoded if adopting a combinatorial strategy of primer selection[34].

The complete encoding and decoding procedures are illustrated in Fig. 2b. The text data must be pre-processed (Supplementary Fig. 7) so that every file in the data is clearly segregated and every prepared data segment contains uninterrupted words to facilitate searching. Additionally, the length of every data strand should lie within the optimal range for PCR amplification and DNA sequencing. Taking the word "CRISPR" as an example, the NCG coding groups data units (Supplementary Fig. 8) representing "CR", "IS", and "PR" into Groups #641, #650, and #716, respectively. The reference pool can be viewed as a two-dimensional (2D) matrix where the reference strands occupy each row while the 16-bit data fragments can be inferred from each column, which refers to the base position. The binary ASCII code of "CR" ("0100001101010010") is mapped to base position #716, which is the same as its group index, with a pointer "G". The base "G" at base position #716 then fills reference strands referring to bits where "1" occupies in ASCII-coded "CR". The determination of pointer follows the order in a defined base mapping matrix (Supplementary Fig. 9). The other two data units are analogously mapped to the reference strands. In practice, each reference strand is an 80-nt piece of the original reference strand to comply with the length requirement for PCR amplification and sequencing. The data strand is constructed by concatenating multiple 7-nt data units in its original order

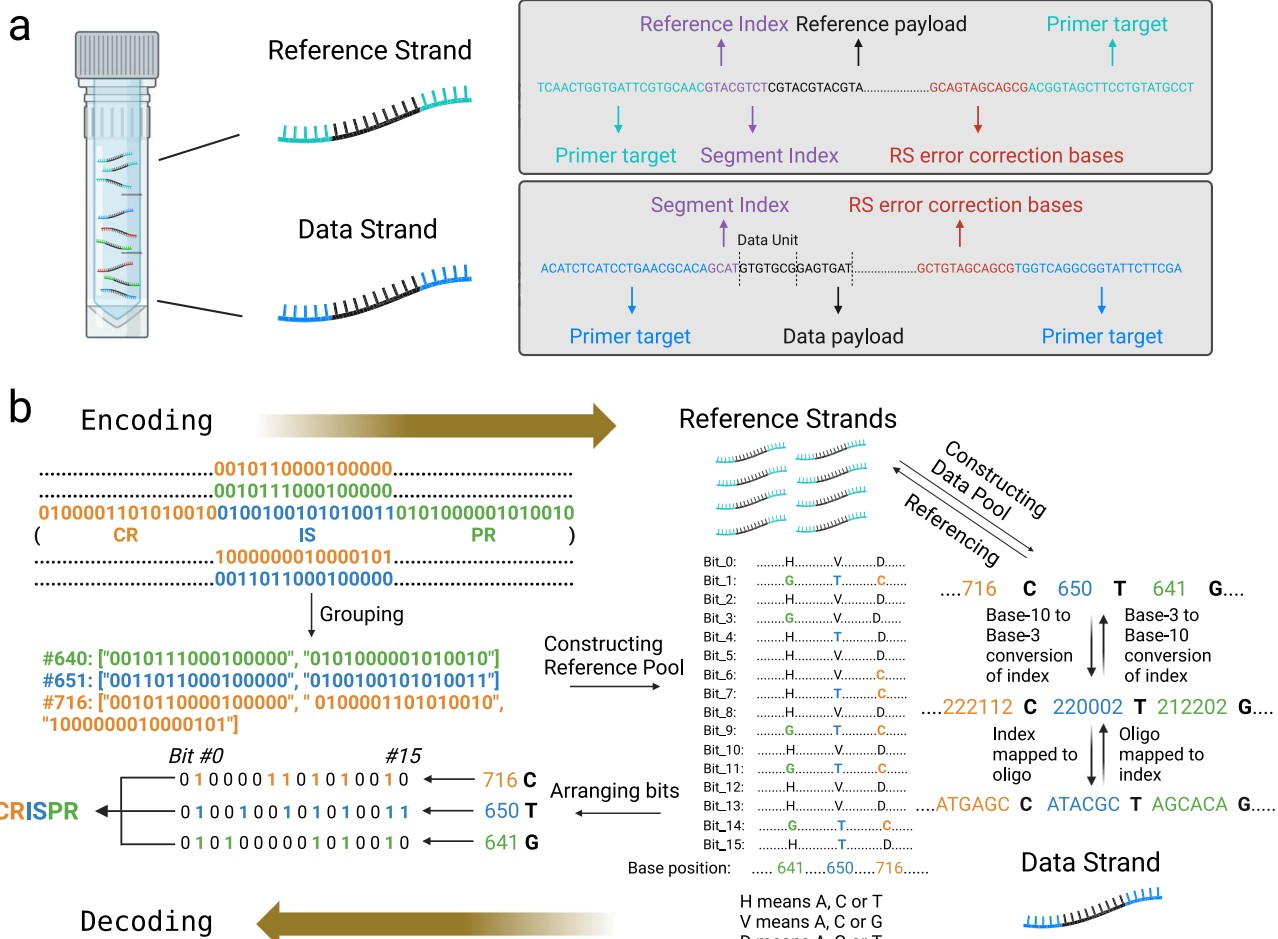

**Fig. 2 | NCG coding procedure and performance. a** Structure of the reference and data strands. A reference strand includes 21-nt forward and reverse primer target sequences, a 4-nt reference index, a 4-nt segment index, an 80-nt reference payload, and 12-nt RS error correction bases. A data strand includes 21-nt forward and reverse primer target sequences, a 4-nt segment index, an 84–112-nt data payload consisting of multiple 7-nt data units, and 12-nt RS error correction bases.
**b** Workflow of the encoding/decoding procedure. Encoding starts from grouping 16-bit data fragments without collision of "1"s at a specific bit position. The groups are assigned an index. The construction of reference strands can be viewed as filling a 2D matrix, where rows represent the bit position and columns represent the group index. The pointer bases filling the reference strands are selected by traversing a pre-defined base mapping matrix. Each column in the reference pool is filled with pointer bases whose row numbers equal to the bit positions of "1" in a data fragment. The payload of a data strand is a concatenation of sequence units comprising a base-3-converted group index and a pointer. Decoding is literally a reverse process of encoding, in which 16-bit data fragments are inferred from the columns of recovered reference pools at corresponding group indexes with a specified pointer as indicated in the data units, converted to text symbols, and concatenated to retrieve the full text. Illustrations were created with BioRender.com.

(Supplementary Fig. 10). The first six bases of a data unit are generated by base-3 conversion of the group index and homopolymer-free base mapping[3] and the last base is the pointer.

Decoding of the data pool should always be accompanied by an already decoded reference pool (Supplementary Figs. 11–12). Since the lengths of the data strands vary and some are longer than the sequencing read length (151 bp), the paired-end reads must be carefully aligned to recover the full-length data strands (Supplementary Fig. 13). Each 7-nt data unit of the data strands is then extracted, of which the first six bases are reversely converted to a group index and the last base is recognized as the pointer (Supplementary Fig. 14). In decoding "CR" as an example, the group index #716 matches the base position #716 in the reference strands. The indexes of the reference strands where the bases at position #716 are identical to the pointer "G" are collected. In a 16-bit data fragment, the bits at these collected indexes are assigned "1" while other bits are assigned "0". After bit assignment, the data fragment can be directly mapped to "CR" according to the ASCII table. The other letter combinations "IS" and "PR" are decoded in the same manner.

## Keyword searching with SEEKER in oligo pools

Since the trans-cleavage activity of Cas12a is indiscriminate, signal interferences from matched DNA strands of unamplified files should be prevented. To achieve this, the fluorescence signal must be hardly detectable when the oligo pool is diluted to a low concentration, while being recovered if a file containing the target is amplified. This condition ensures that the signal produced from trans-cleavage is specific to the file being searched. As a preliminary experiment to examine the viability of this condition, we designed two oligo pools respectively encoding File A and File B (Supplementary Data 1). Each oligo pool consisted of only ~50 oligos so that a relatively high yield (~50 pmol/oligo) could be obtained. We used a 10× dilution of the original oligo pool such that each oligo had a sub-nanomolar concentration. We were able to correctly identify the oligo pool containing the keyword within 120 min (Fig. 3a). Enhancing the concentration of the ssDNA-FQ reporters produced stronger fluorescence on the target files to facilitate visualization of results, and also magnified the gap of fluorescence intensity between target and non-target files to make identification clearer (Supplementary Fig. 15). Due to baseline

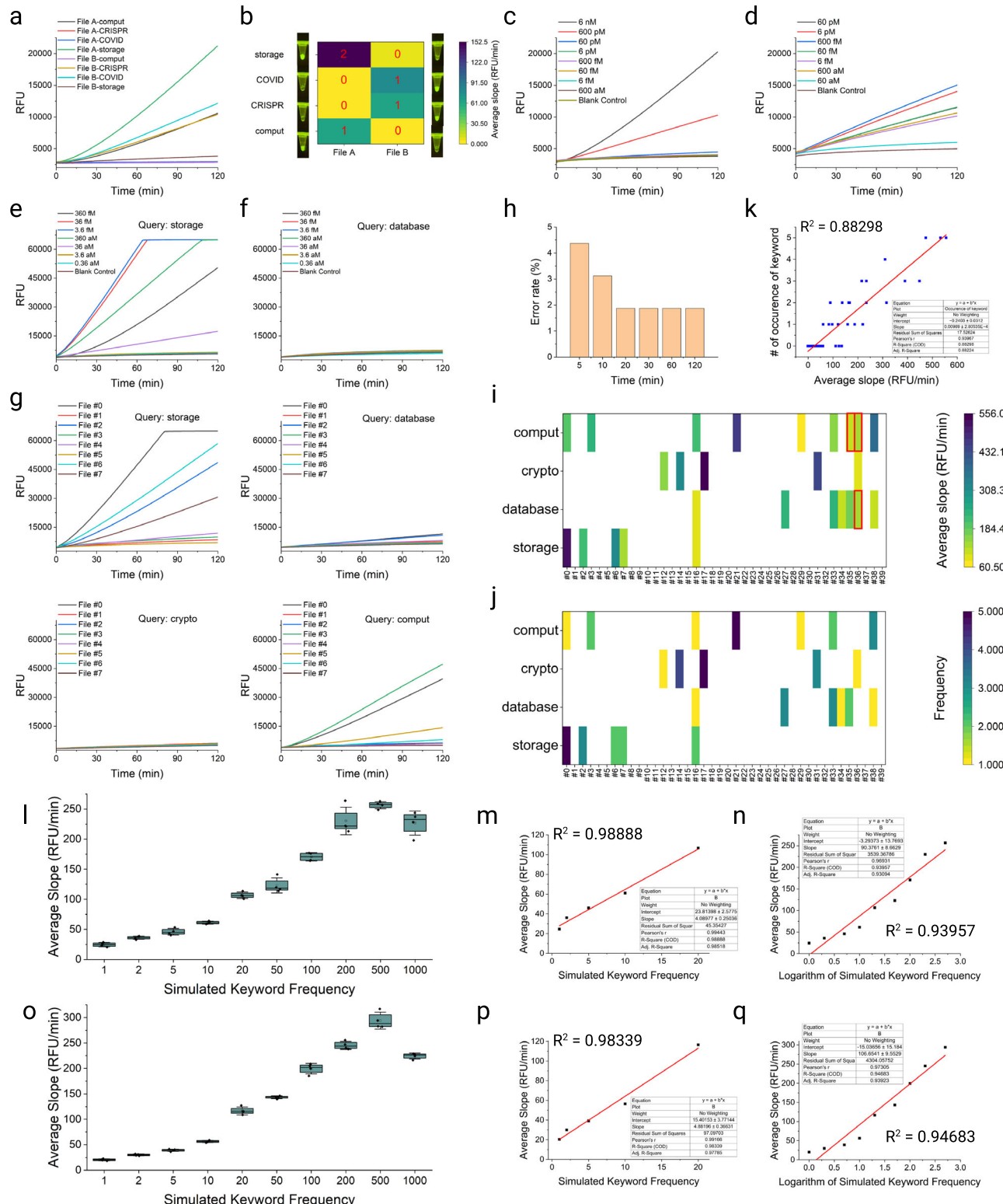

variations among experiments, we used the slope, which is the average change in fluorescence intensity per minute from the start of the reaction, rather than the absolute fluorescence intensity to identify target and non-target files. We found that the slope of fluorescence intensity was proportional to the frequency of the keyword in the file being searched, suggesting this method can be quantitative in keyword searching (Fig. 3b).

We then investigated the limit of detection in a single-file oligo pool. For diluted oligo pools without amplification, we only observed

significant gaps of fluorescence intensity for concentrations above 60 pM (Fig. 3c). We then performed PCR with primer sequences matching the first and last 21 nt of Strand #2 of File B which contained the query sequence exactly. We selected the template concentrations from those not detectable using unamplified oligo pools. When we applied the PCR amplicons to the SEEKER system, the fluorescence intensities for template concentrations as low as 600 aM were fully restored to the same level as when using an unamplified high-concentration oligo pool (Fig. 3d). These observations on a single-file

**Fig. 3 | Performance of SEEKER in single-file and multi-file oligo pools. a** Real-time fluorescence kinetics of keyword searches in unamplified oligo pools at a final concentration of 600 pM. Four queries ("comput", "CRISPR", "COVID", and "storage") were used to search File A and File B. **b** Average slopes of fluorescence at 120 min for all searches in (**a**). Readouts under blue LED are shown in corresponding side images. Red numbers refer to the frequency of a keyword. **c, d** Real-time fluorescence kinetics of a search in an oligo pool referring to File B by the query "COVID" without amplification (**c**) and with Strand #2, which contained the keyword "COVID" amplified (**d**).Template concentrations are specified in the legend. **e, f** Real-time fluorescence kinetics of searching by the queries "storage" (**e**) and "database" (**f**) in amplified File #0. The oligo pool was serially diluted before performing PCR to determine the minimum template concentration required for SEEKER. **g** Real-time fluorescence kinetics of searching in amplified Files #0–#7 using four keywords ("storage", "database", "crypto", and "comput"). **h** Error rates in target file identification across different reaction times. **i** Average slopes of fluorescence for all searches in 40 files over 20 min. Red boxes indicate misidentified files under specific queries. **j** Frequency of each keyword in every encoded file. The limit of keyword count is 5. **k** Linear correlation between the average slope of fluorescence and the frequency of the keyword. **l** The average slope of fluorescence with different keyword frequencies simulated with 1-keyword oligos ($n = 4$). The center line represents the median value, the bounds of the box indicate the interquartile range, and the whiskers indicate mean ± SD of four technical replicates). **m** Linear fitting between average slopes and simulated keyword frequencies using 1-keyword oligos when keyword frequency was below 20. **n** Linear fitting between average slopes and logarithms of simulated keyword frequencies with base 10 using 1-keyword oligos when keyword frequency was below 500. **o** The average slope of fluorescence with different keyword frequencies simulated with a mixture of 1-keyword, 2-keyword and 5-keyword oligos ($n = 4$. The center line represents the median value, the bounds of the box indicate the interquartile range, and the whiskers indicate mean ± SD of four technical replicates). **p** Linear fitting between average slopes and simulated keyword frequencies using multi-keyword oligos when keyword frequency was below 20. **q** Linear fitting between average slopes and logarithms of simulated keyword frequencies with base 10 using multi-keyword oligos when keyword frequency was below 500. For **l–q** the slopes represent the average fluorescence enhancement in the first 30 min of reaction. Source data are provided as a Source Data file.

oligo pool provided the basis for applying SEEKER to more complex multi-file cases, where random access is allowed but should not interfere with the search results.

We then applied NCG coding to a plain text containing abstracts of 40 randomly selected journal articles (Supplementary Data 2). To demonstrate random access to files stored in DNA, we encoded each abstract as a single file in the data pool. After pre-processing, these 40 files yielded 717 groups, close to the defined encoding volume, which was 729. In practical scenarios, we suggest the number of groups as close as possible to the defined encoding volume for a higher coding potential. The size of the original data was 43.544 KB. After NCG coding, there were 144 oligos in the reference pool and 1519 oligos in the data pool. The coding potential was 2.05 bits/nt, suggesting that the original data was effectively compressed by NCG coding. When we included nucleotides for primer IDs, segment indexing, and RS error corrections in each strand, the net information density was 1.33 bits/nt.

The yield decreased as the oligo pool became more complex. We dissolved the oligo pool to 6 pM in estimation. This concentration level, along with its serial dilutions, was unlikely to produce a recognizable fluorescence readout using an arbitrary query, for instance, the word "storage" which was present in the oligo pool (Supplementary Fig. 16). When using the same query to search a selectively amplified File #0, which contained this query, the fluorescence response became strong and distinguishable with varying template concentrations (Fig. 3e). When using the query "database" which was absent in the amplified File #0, we observed no obvious fluorescence response across the same template concentration range (Fig. 3f). Since this assay is quantitative, the fluorescence intensity for an amplified file that contains more copies of the keyword may be higher. We then optimized the template concentration using multiple files with a lower keyword frequency and determined an optimal template concentration of $10^{-1}\times$ of the original concentration, which is equal to 36 fM or $2.17 \times 10^4$ copies/μL, as the concentration for the following experiments (Supplementary Fig. 17).

To demonstrate the feasibility of SEEKER in large-scale applications, we selected four queries, "storage", "database", "crypto", and "comput", to search all 40 files encoded in the oligo pool. Figure 3g shows the real-time fluorescence kinetics for searching Files #0–#7 within 120 min. For each query, we calculated the average slopes of the fluorescence enhancement for all 40 encoded files. We set the threshold as the minimum slope observed for a target file and regarded non-target files with slopes greater than the threshold as misidentified files. Thereby, we obtained an error rate by dividing the number of misidentification files by the total number of searches. The error rate of SEEKER declined as the reaction time prolonged in the first 20 min, after which the error rate became stable (Fig. 3h and

Supplementary Fig. 18). After 20 min of reaction, we observed three misidentified files among 160 searches with an error rate of 1.875% (Fig. 3i). Despite the fact that this obtained error rate is higher than that in a digital computer system, it has been theoretically and experimentally proven to be superior to the conventional hybridization-based approach (Supplementary Fig. 19, Supplementary Table 6), where targets with mismatches on several continuous bases may exhibit even stronger binding affinity than the perfectly matched one. We speculate that the improvement on identification accuracy can only be achieved by a more stringent criteria in nucleic acid sequence design, ensuring minimal similarity and enough orthogonality of sequences encoding unique keyword and features.

To determine whether using different Cas12a nucleases might impact the performance of SEEKER, we tested another commercially available Cas12a ortholog EnGen® Lba Cas12a and observed a consistent result of keyword searching (Supplementary Fig. 20). We then counted the frequency of each keyword in every encoded file (Fig. 3j) and confirmed the average slope of the fluorescence enhancement was linearly correlated with the frequency of the keyword, with an $R^2$ value around 0.88 (Fig. 3k). We further designed experiments to investigate the performance of SEEKER with higher keyword frequencies up to 1000 by adjusting the proportion of keyword-containing oligos in the entire oligo pool. The keyword frequency was simulated with oligos purely containing one keyword in each strand (Fig. 3l) and a mixture of oligos containing one, two or five identical keywords in each strand (Fig. 3o, Supplementary Tables 7, 8) to better approach the practical situation where keywords are repeatedly used in a sentence. We observed perfect linearity between keyword frequency and fluorescence slope in both conditions when keyword frequency was below 20 (Fig. 3m, p). When the keyword frequency further increased, the linearity was compromised but could still result in an $R^2$ greater than 0.8 when the frequency was below 200 (Supplementary Fig. 21). Furthermore, we noticed that with high keyword frequencies, the quantitative relationship became more logarithmic rather than linear, and a good logarithmic relationship was obtained when keyword frequency reached 500 in both experimental conditions (Fig. 3n, q). These results validate that SEEKER is linearly quantitative in searching text data stored in DNA when keyword count is below a certain limit, and is logarithmically quantitative with high keyword frequencies if not reaching the saturation level. Assuming the relevance of a keyword to an article is often represented by how many times the keyword is repeated, SEEKER is able to return results with a "relevance level", which may further guide readers on which file to read.

Moreover, we demonstrated that the amplified oligo pool used in a previous SEEKER reaction can be reused in sequential new SEEKER reactions. Although it is commonly believed that the CRISPR-Cas12a

system will cleave the double-stranded DNA targets, effective cleavage only occurs with sequences containing the canonical protospacer adjacent motif (PAM)[35]. In our sequence design, no canonical PAMs were included in the data strands. We speculate that the amplified oligos, despite containing the sequence encoding the keyword, will remain undamaged after one SEEKER reaction and can be reused. To simplify the experimental procedure, the cleaved fluorescence probes were not removed from the last reaction. However, it is necessary to clear the last crRNA query as it may cause interference in a new search, which can be achieved by the addition of RNase I$_f$. The experimental results (Supplementary Fig. 22) indicated that SEEKER can be used for at least three sequential rounds, despite some differences observed in the real-time fluorescence kinetics which could possibly be attributed to the addition of RNase I$_f$. Moreover, we found the sample volume from the last reaction appeared to have a minimal effect on the fluorescence enhancement rate.

## SEEKER on a chip

Considering the potential immense applications of DNA data storage, we intended to design SEEKER as a convenient tool for data searching without burdensome labor, complicated procedures, or rigorous experimental conditions. The "lab-on-chip" concept motivated us to perform SEEKER on a 3D-printed chip, which is made of resin and can be mass-produced at a low cost. The disc-shaped chip (Fig. 4a), with a diameter of 45 mm and a thickness of 4 mm, consisted of arrayed sample-loading chambers with channels connecting them. The central chamber on the chip is primarily used to introduce the CRISPR reaction mixture, which then flows through microchannels and fills up each side chamber. The side chambers are the inlets for amplicons corresponding to a file being searched. The chip allows 20 files to be searched simultaneously. The middle chambers are created to prevent resin from clogging in the microchannels during stereolithography-based fabrication. Before use, we pushed the CRISPR reaction mixture

from the central chamber to allow it to flow through the channels and fill up the surrounding side chambers. The CRISPR mixture can also be supplied from the middle chambers to fill up the side chambers if the liquid distribution is uneven. The chip can then be lyophilized for long-term storage[36,37].

When in use, the chip was rehydrated with crRNA solution introduced from the central chamber. Amplified DNA files were then mixed with the solution in the side chambers to initiate the reaction. Due to the small volume of amplicons used, after gentle mixing the reaction mostly occurred in a zone called the detection channel, which connects the side chamber and a nearby middle chamber. Variations in the fluorescence intensities between target and non-target files became apparent within 20 min and were visually detectable (Fig. 4b). We achieved an accurate measurement of fluorescence intensity by quantitatively analyzing the grayscale intensity of the detection channel. We conducted on-chip SEEKER using two queries ("database" and "crypto") and defined the threshold as the lowest grayscale intensity of a target file. We observed the same misidentification of File #36 using the query "database" as we did in the in-tube experiments, while all other search results were correct (Fig. 4c). In addition, linear regression between the grayscale intensity and frequency of keywords with an R$^2$ of 0.71 (Fig. 4d) suggests that the on-chip SEEKER is capable of conducting quantitative searching. Overall, the performance of on-chip SEEKER was comparable with that of in-tube SEEKER.

## Reading NCG-coded files through sequencing

We mixed amplified target files corresponding to the four queries and performed Illumina MiSeq sequencing together with the amplified reference pool to demonstrate whether DNA data encoded through NCG coding can be fully retrieved. With an average sequencing coverage of ~40× (Supplementary Table 3), we were able to recover the entire amplicon library of the data pool. We found the reference pools were more prone to base calling errors because the reference strands

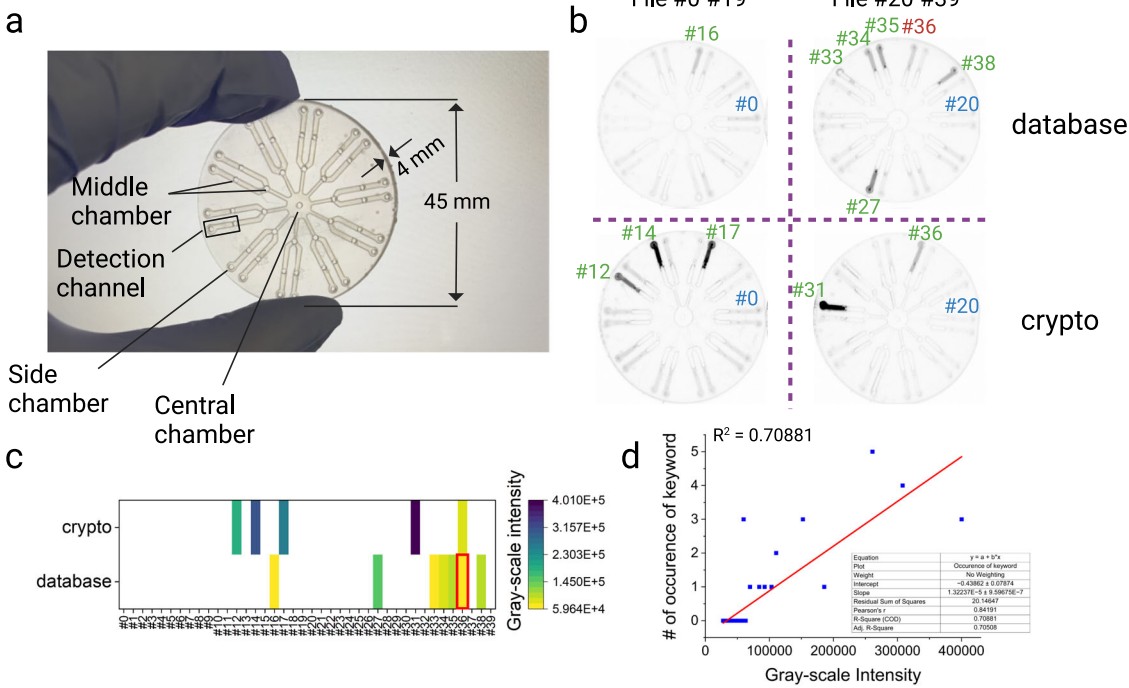

**Fig. 4 | SEEKER on a chip. a** Photograph of a 3D-printed microfluidic chip implementing SEEKER. **b** Fluorescence readout of searching results after 20 min of reaction. Blue numbers mark the start of file loading on the chip. Files were loaded clockwise following their original order. Green numbers refer to correctly identified files and red numbers refer to misidentified files. **c** Grayscale intensity of every detection channel imaged in (**b**). Red box indicates misidentified files under specific queries. **d** Linear correlation between the grayscale intensity of the detection channel and the frequency of the keyword. Source data are provided as a Source Data file.

may include cycles of "ACGT" or its shifted permutations "CGTA", "GTAC", or "TACG" (Supplementary Table 4). However, incorrect base calls can be remediated through an algorithm provided in Supplementary Fig. 12 so that the reference pool can be fully recovered. The recovered sequences can be directly converted back to readable texts using the source code provided.

Bias in PCR amplification, which can be possibly caused by improper primer design or difficulty in amplification of certain payload sequences, can result in missing files even if high sequencing coverage is applied. Ideally, if no bias exists in amplification, the possibility of obtaining valid reads from any file of interest should be equal, as each amplified file contributes an equal volume to the entire amplicon pool in the experiment. Counting valid reads from every file of interest in the four reconstituted amplicon pools of data strands (Fig. 5a) showed that the largest deviation between the actual and ideal proportion of reads from a specific amplified file was 10.31% (File #16 in data #1), with an average deviation of 4.904% for data #1, 1.718% for data #2, 5.576%

for data #3, and 2.03% for data #4, indicating the chances of a significant bias to a specific file during amplification were very low. Furthermore, an in-depth analysis revealed that, on average, 73.57% of valid reads after sequence alignment in the decoding process exactly matched sequences from the data pool and the reference pool (Supplementary Table 5). RS correction code improved this ratio by 5.58% on average (Fig. 5b), leading to a total of ~2000 fully recovered reads that were originally undecodable with minor base calling errors.

To investigate how a lower sequencing coverage may impact the recovery of NCG-encoded data, we randomly subsampled the original reads and calculated the dropout rate, which is defined as the proportion of the number of missed amplicon species to the total number of amplicon species in a subsampling experiment. In addition, we defined "full access" as when the dropout rate is zero and defined "full access rate" as the proportion of times of full access to the total number of subsampling experiments, which was 20 in our study. For all five samples, we found that the estimated sequencing coverage ranged

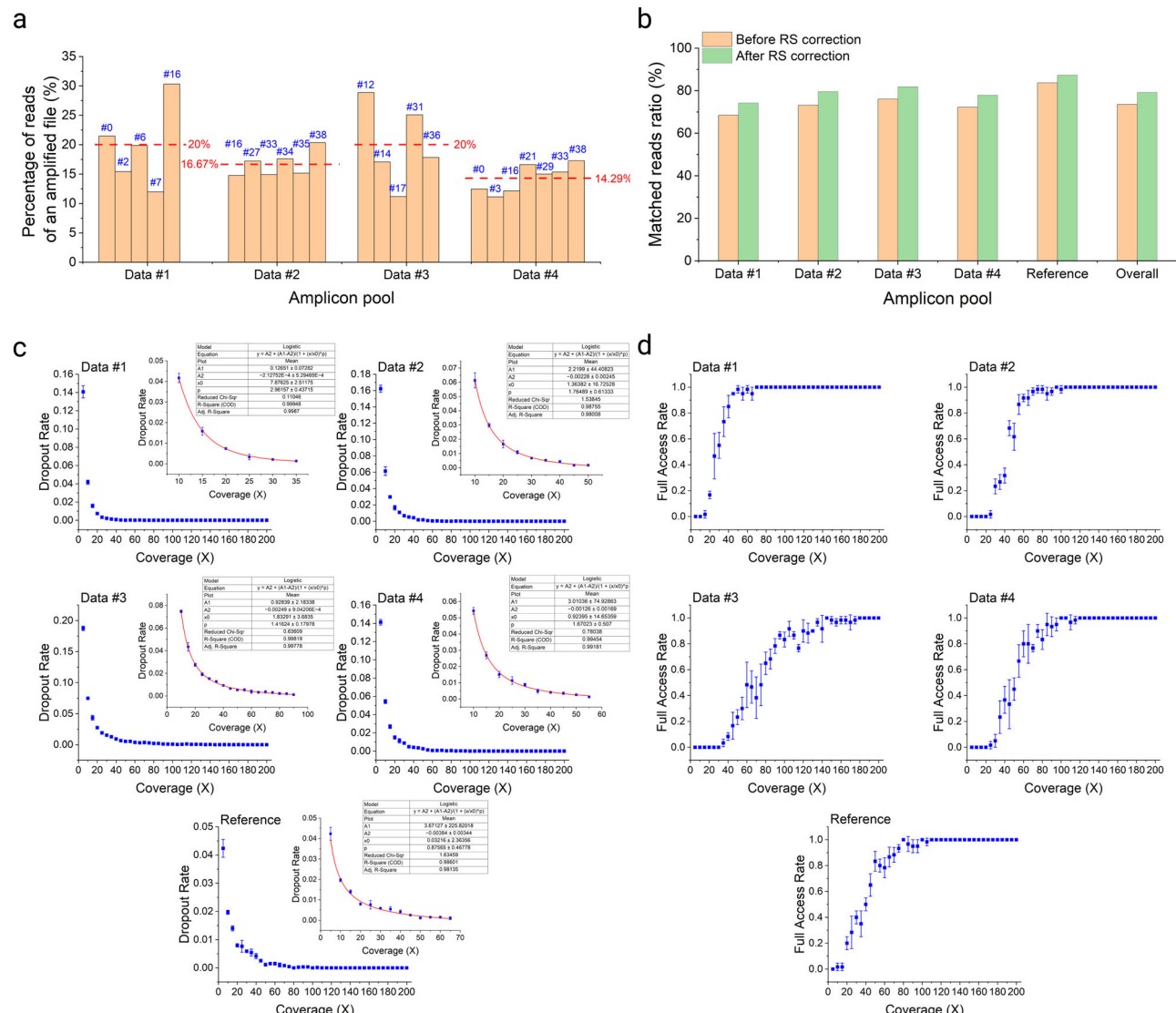

**Fig. 5 | Reading NCG-coded files through sequencing. a** Fractions of reads from each file of interest in the corresponding amplicon pool. The expected average fractions are shown as red dashed lines marked with percentages. **b** Comparison between the ratios of matched reads before and after RS correction. **c** Dropout rates of subsampled reads in expanded FASTQ files. The sample names are specified in the figure. The insets show zoomed-in views when the dropout rates are close to 1%. For the insets, logistic regressions were performed to study the

relationship between sequencing coverage and dropout rates as well as to estimate the minimum coverage to achieve dropout rates less than 1%. **d** Full access rates of subsampled reads in expanded FASTQ files. The sample names are specified in the figure. For (**c**, **d**) the sequencing coverage ranged from 5× to 200× with an interval of 5× and the dropout rate at a specific coverage was the average of results from three repeated experiments of 20 independent subsamplings. Data are presented as mean ± SD. Source data are provided as a Source Data file.

10× to 35× when the cutoff dropout rate was 1% (Supplementary Fig. 23). However, except for data #1, the full access rate dropped drastically as the sequencing coverage slightly decreased (Supplementary Fig. 24), potentially suggesting the coverage before subsampling already approached the minimum coverage needed for full recovery. To estimate the minimum coverage more accurately, we expanded the sample size by resampling the original reads to ~200× of the total number of amplicon species and performed random subsampling on the enlarged sample. When decoding the resampled reads, the coverages required to reach a dropout rate of 1% were 17.91×, 25.86×, 38.10×, 26.17×, and 18.78× for data #1, data #2, data #3, data #4, and the reference pools, respectively (Fig. 5c). The minimum coverage was typically less than 40×, which is consistent with the results obtained from the raw reads. In addition, the minimum coverages when all three individual experiments showed 20 full access out of 20 subsamplings were 70×, 95×, 135×, 100×, and 80× for data #1, data #2, data #3, data #4, and the reference, respectively, with an average of 96×. The minimum coverages needed to obtain at least one full access out of 20 subsamplings for all three individual experiments were 20×, 30×, 40×, 35×, and 20× for data #1, data #2, data #3, data #4 and the reference, respectively, with an average of 29× (Fig. 5d). These results indicate that NCG-coded DNA data can be fully recovered at a regular sequencing coverage within the range of those commonly reported[3,4,6-8,16,38].

## Discussion

Convenient and reliable content-based searching of data stored in DNA is one of the fundamental challenges that must be addressed before DNA data storage can be broadly applied in reality. SEEKER provides a solution to implement keyword searching on text data stored in DNA. This system utilizes the trans-cleavage activity of CRISPR-Cas12a, expanding its applications from widely known molecular diagnostics to DNA data storage and potentially many other new directions. Through real-time measurement of fluorescence intensity, files containing the keyword can be identified within minutes, and the fluorescence enhancement is numerically proportional to the frequency of the keyword, rendering SEEKER both qualitative and quantitative for keyword searching. We project that "searching-while-computing" will become a crucial concept in guiding the future development of DNA data storage systems since, in this data explosion era, people tend to dismiss irrelevant information and expect to efficiently look up information more correlated to their interests, and the relevance should be computable during the search process.

Detection with the CRISPR-Cas12a system relies on a protospacer adjacent motif (PAM) on the DNA strand opposite to the target sequence[39,40]. While this property has been widely recognized, we found that PAM was not necessary for SEEKER, as we designed all data strands without the incorporation of PAM in any targets and still achieved successful detection. As PAM is intended to unwind the double strand and allow binding to crRNA, we assume that it is only mandatory when the target is in a double-stranded form. Although the theoretical DNA products of PCR are double-stranded, some may remain single-stranded because of unsuccessful primer binding or random termination of the reaction before primer extension, and these single-stranded targets can be detected without PAM restrictions. Besides, recent study suggests non-canonical PAM is also capable of initiating the trans-cleavage reaction[35]. However, we note that the increase of fluorescence in our study was slower than other reported studies when the target was at a similar concentration level[30,41], which may be attributed to the lack of PAM.

Compared to hybridization-based approaches[17,18], the CRISPR system is believed to have higher specificity in target recognition[42,43] and is anticipated to implement searches with higher accuracy, especially in a noisy background where there is a high probability of capturing a random sequence similar to the target. Nonetheless,

misidentification can still occur, as in this study 3 out of 160 searches produced false-positive results. We excluded the possibility that improper design of PCR primers induced non-specific amplification, as we found no sequences belonging to irrelevant files in the sequencing reads of Files #35 and #36 where mistakes occurred. We next reasoned that the non-specificity of the CRISPR reaction caused the misidentification. We performed Needleman–Wunsch alignment[44] to analyze the similarity between every 21-nt sequence fragment of data strand in each file (including primer sequences) and query sequences encoding "database" and "comput" where incorrect identifications appeared. We listed all sequence fragments that had a sum of gap penalty and mismatch penalty smaller than 5 as "highly similar" sequences to the query sequence (Supplementary Tables 9–11). We marked in red highly similar sequences uniquely found in File #35 and #36 but not observed in other files. These marked sequences are then assumed to be the interfering targets falsely recognized in the CRISPR reaction.

Quantitative PCR (qPCR), another technique for molecular diagnostics, presumably achieves the same function since it can be designed with a sequence-specific DNA probe that hybridizes with its complementary target sequence as crRNA does. However, CRISPR is superior to qPCR in the following aspects: (1) Experimental conditions. Every search attempt with qPCR requires a high temperature (95 °C) and multiple thermocycling procedures, which can only be achieved with bulky equipment, thus preventing this method from being implemented in a regular setting. (2) Time. Fluorescence in qPCR can only be revealed after many cycles, while the trans-cleavage of CRISPR-Cas12a is triggered immediately upon the addition of amplicons. (3) Material consumption. Every qPCR process consumes the oligo pool or its dilution as the template. By contrast, once the DNA file has been amplified, the amplicons can be used in SEEKER multiple times because every experiment consumes only 1.5 μL of the amplicon pool, making the searching process more cost-effective. Furthermore, recombinase polymerase amplification[45], an isothermal alternative to PCR that can be operated at 37–42 °C, allows amplification and detection of SEEKER implemented in one step[26,30] and is fully operable in ambient conditions.

In our study, some sequences in the reference pool contained short four-base repeats that were prone to sequencing errors[46]. The major reason for the occurrence of repeats is that the generation of the reference strand through base mapping matrix (Supplementary Fig. 9) strictly followed an order starting from groups of one combination, and then two and three combinations. Before the remedial process, the summary of missing sequences of the reference pool after sequencing (Supplementary Table 4) revealed that all undecodable sequences solely contained repeats with up to one single-base permutation of the four-base repeat. This suggests that the sequencing error can be avoided by rearranging the order of elements in groups, with groups of one combination intercalated with groups of multiple combinations, to ensure that each reference strand has two or more permutations of the repeats. This approach is feasible in our study since there were 77 groups of multiple combinations out of 717 groups in total, enabling an average of 8.6 permutations inserted in each 80-nt reference strand. Alternatively, sequencing errors can be reduced simply through randomly changing the base order in base mapping at specific sites of the reference strands, at the expense of extra memory to record the changes.

SEEKER can also be adapted to more advanced and practical search schemes, such as a metadata search, which only requires a slight modification to the oligo data structure (Supplementary Fig. 2a). By combining microfluidics as a DNA storage medium, SEEKER is able to determine the physical location of the file containing the keyword through visible and digital fluorescence states (0/1), eliminating the need for isolation and sequencing of every chamber. Acknowledging the limitation of SEEKER in physically isolating strands carrying the

keyword or features being searched, we believe a more effective approach is to combine SEEKER with conventional hybridization-based methods (Supplementary Fig. 2b). SEEKER could serve as a pre-screening routine to rapidly identify whether the keyword or feature exists in the enriched oligo pool. Subsequently, a hybridization-based approach can be implemented to isolate oligos from the candidate pools and read the metadata through sequencing. This way, both methods can complement each other to make molecular searching in DNA databases more convenient and efficient.

## Methods

### Data encoding and decoding
A step-by-step data encoding and decoding protocol is available in the flowcharts in Supplementary Figs. 7–12, 14. The complete encoding and decoding pipeline constitutes: (1) pre-processing of text data; (2) searching groups without position collisions through the NCG algorithm; (3) generating reference sequences; (4) generating data sequences; (5) decoding reference sequences (including the remedial process); and (6) decoding data sequences. The program[47] is written in Python 3 and the code is available on GitHub at https://github.com/Jiozhang/SEEKER-encoding-and-decoding.

### Primer design
A detailed protocol of the PCR primer design is provided in Supplementary Fig. 5. In the initial screening, only primers with balanced GC content and absence of homopolymers are preserved to reduce the error rate in DNA sequencing. In addition, the PCR primer sequences should not be identical or very similar to query sequences or their reverse complementarities, as interferences will occur when CRISPR misrecognizes abundant PCR primers or amplicons containing the reverse complementary sequences of PCR primers as targets. Next, the primer candidates are subjected to a series of thermodynamic and orthogonality screenings. Thermodynamic screening eliminates primer candidates that form heterodimers, homodimers, or hairpin structures at the annealing temperature of PCR. Meanwhile, the melting temperature of each primer candidate should be kept between 60 °C and 65 °C, slightly higher than the annealing temperature used in the PCR protocol. Orthogonality screening reduces crosstalk to ensure the amplification is specific to an intended file rather than irrelevant files. To be specific, the orthogonality screening should ensure that the Hamming distance between any two primer candidates is smaller than 6, and there is no more than 10 bases of sequence complementarity between any two primer candidates[16]. We have also screened the Illumina overhang adapter forward sequence 5'-TCGTCGGCAGCGTCAGATGTGTATAAGAGA-CAG-3', reverse sequence 5'-GTCTCGTGGGCTCGGAGATGTGTATAA-GAGACAG-3', and their complementarities and found no overlap with the query sequences and primer sequences. To facilitate sequencing, the selected primers were prepended with Illumina adapter sequences at the 5' end before ordering.

### File selection through PCR
All primers and the single-file oligo pool were ordered from Integrated DNA Technologies (IDT). The multi-file oligo pool was ordered from Twist Bioscience. The sequences of primers and oligos in single-file and multi-file oligo pools are provided in Supplementary Data 3–6. All oligos were dissolved in nuclease-free water (New England Biolabs, Cat# B1500S). To perform selective amplification of the DNA files, 12.5 μL of 2× KAPA HiFi HotStart ReadyMix (Roche Diagnostics, Cat# 50–196-5217), 1 μL of 100 μM forward primer, 1 μL of 100 μM reverse primer, 1.5 μL of oligo pool template, and 9 μL of nuclease-free water were mixed in PCR tubes to make a 25 μL reaction. The PCR was performed in a CFX96 Touch Real-Time PCR Detection System (Bio-Rad Laboratories) with the following protocol: (1) initial denaturation at 95 °C for 3 min, (2) denaturation at 95 °C for 30 s, (3) annealing at 55 °C for 30 s, (4) extension at 72 °C for 30 s, with steps 2–4 repeated for 35 cycles, and then (5) final extension at 72 °C for 5 min. The amplicons were stored at −20 °C before use.

### In-tube SEEKER
The ssDNA-FQ reporters (5'-FAM-TTATT-IABkFQ-3') and crRNAs were ordered from IDT and dissolved in nuclease-free water (New England Biolabs, Cat# B1500S). The sequences of crRNA queries can be found in Supplementary Data 7. In this work, SEEKER was implemented with dual crRNAs since the encoding of one word can either start from a letter or from a whitespace word divider. The letter case should also be considered in practice as it affects the query sequence. Each in-tube SEEKER was a 25 μL reaction composed of 2.5 μL of 10× NEBuffer™ r3.1 (New England Biolabs, Cat# B6003S), 2.5 μL of 50 μM ssDNA-FQ, 1.5 μL of 5 μM Alt-R® A.s. Cas12a (Cpf1) V3 (Integrated DNA Technologies, Cat# 1081068), 3 μL of 100 nM crRNA Query #1, 3 μL of 100 nM crRNA Query #2, 1.5 μL of PCR amplicons, and 11 μL of nuclease-free water. The addition of PCR amplicons was performed in a PCR workstation (AirClean Systems) to avoid cross-contamination. The reactions were incubated in a CFX96 Touch Real-Time PCR Detection System (Bio-Rad Laboratories) at 37 °C for 2 h with fluorescence recording every 15 s.

### Chip fabrication
The chip was blueprinted on SOLIDWORKS and fabricated by a high-resolution stereolithographic laser-based 3D printer (Form 3B + ) from Formlabs. After completion of printing, the chips were immersed in isopropanol (Fisher Scientific, Cat# 383910025) and subjected to a 25-min ultrasonic cleaning. The chips were then washed with deionized water, air-dried, and stored in a cool and dry place before use.

### On-chip SEEKER
The CRISPR mixture containing NEBuffer™ r3.1 (New England Biolabs, Cat# B6003S), ssDNA-FQ, Alt-R® A.s. Cas12a (Cpf1) V3 (Integrated DNA Technologies, Cat# 1081068) with the same concentration as used for in-tube reactions was prepared and loaded into the central chamber. A minimum of a 350 μL mixture was required to fill up every side chamber on the chip. The chips were then flash frozen in a −80 °C freezer and subjected to lyophilization overnight. The lyophilized chips were stored in a sealed plastic bag at 4 °C before use. When implementing the on-chip SEEKER, 100 nM of the crRNA queries were introduced from the central chamber until each side chamber was filled up. Then, 1.5 μL of amplicons corresponding to the files being searched were loaded into each side chamber and mixed gently by pipetting up and down. The chips were then sealed with PCR tape (Thermo Fisher Scientific) and placed in an incubator (Thermo Fisher Scientific) set at 37 °C for 20 min. The fluorescence readouts on the detection channel were imaged by the ChemiDoc MP Imaging System (Bio-Rad Laboratories) and their grayscale intensities were analyzed through ImageJ.

### Sequencing
A sequencing sample was a mixture of amplicons meant to carry a specific keyword. The library preparation followed the 16 S Metagenomic Sequencing Library Preparation Workflow. In detail, the amplicons containing Illumina sequencing adapters were purified using AMPure XP beads (Beckman Coulter Genomics, Cat# A63881), and then Index PCR was performed using the Nextera XT Index Kit (Illumina, Cat# FC-131-1001) to attach dual indices. A second PCR clean-up using AMPure XP beads was performed to obtain the final library. The final library was diluted to 4 nM using 10 mM Tris (pH 8.5) (Thermo Fisher Scientific, Cat# J61038.AP), denatured with NaOH (Fisher Scientific, Cat# SS274-1) and then diluted with hybridization buffer (Illumina, Cat# 20015892) before MiSeq sequencing. The library was loaded into a MiSeq reagent cartridge at 8 pM with a PhiX control (Illumina, Cat# FC-110-3001) spike-in of ≥10%.

## Reporting summary

Further information on research design is available in the Nature Portfolio Reporting Summary linked to this article.

## Data availability

The source data underlying Figs. 3–5, Supplementary Figs. 3–4, 6, and 15–24 in this study are provided in the Source Data file. Matlab R2022b was used to analyze results and produce Supplementary Fig. 3a–b and Supplementary Fig. 4d. All other figures showing experimental or analytical results were produced with OriginPro 2022b. The raw sequencing reads of the reference pool and data pools #1–4 in FASTQ format are publicly available in "Raw reads.zip" file at https://github.com/Jiozhang/SEEKER-encoding-and-decoding. Source data are provided with this paper.

## Code availability

The source code implementing the entire procedure from data pre-processing to encoding and decoding, as illustrated in Supplementary Figs. 5, 7–12, and 14, was written in Python 3 and is publicly available on GitHub at https://github.com/Jiozhang/SEEKER-encoding-and-decoding. The primary packages required to implement the code include primer3 and reedsolo. All packages and dependencies required to execute the code are listed in "requirements.txt". Examples used in this work are provided along with the code to assess reproducibility.

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

## Acknowledgements

We thank Ms Bo Reese from Center for Genome Innovation, University of Connecticut for her help with amplicon sequencing. The work was supported by startup funds (to C.L.) from the University of Connecticut Health Center. The figures in this work were created with BioRender.com.

## Author contributions

J.Z. and C.L. designed the study. J.Z. and C.H. performed experiments. J.Z. analyzed the data and drafted the paper. C.L. supervised the whole project and edited the manuscript. All authors reviewed and approved the paper.

## Competing interests

The authors declare no competing interests.

## Additional information

**Supplementary information** The online version contains Supplementary Material available at https://doi.org/10.1038/s41467-024-46767-x.

