## [Peer Review File · Nature Communications]

Reviewers' Comments:

Reviewer #1:

Remarks to the Author:

This is interdisciplinary work spanning computer system design and synthetic biology. And it's important to evaluate the novelty and value of the work from both perspectives. My expertise falls mostly on the computer systems side but I am also an expert in DNA-based storage and very familiar with the synthetic biology principles and protocols used in such systems.

Molecular search of data in a DNA storage system is a topic that has garnered attention recently and will likely continue to be of interest. Dense molecular storage systems, if scaled up to a cloud scale of exabytes and beyond, will eventually suffer an extreme IO bottleneck. The time it takes to access and sequence data will be very high limiting what can be sequenced to a small fraction of stored data at a time. If little is known about the data, then it would require prohibitive time and cost to first re-digitize it and then query it. Hence, efforts to directly search data molecularly are worth investigating. For these reasons, the paper would be of interest to a broad community of researchers across the computer system and synthetic biology spectrum. However, it's also important for the approaches explored in this area to build on the best approaches from each domain. I have significant concerns about aspects of this work from the computer systems perspective.

I have two major concerns with this work. (1) The search approach described is far too inefficient to be a useful search primitive in future scaled-out systems. It's only usable at a small scale where digital approaches work well. (2) The proposed NCG algorithm is not demonstrated to have novelty or to benefit the proposed system.

(1) Assuming the high latency and low bandwidth sequencing platforms we have today, the problem of search exists in an extreme form for DNA-based storage. But, this problem already exists today. When a search query, using keywords, is sent to Google, Google doesn't look in every file on the internet to find the answer. Instead, it looks up meta-data that stores information about where words and terms appear and it consults a pre-computed answer for each term and then combines answers along with ranking to return a list of possible matches. This can be done rapidly, accessing very little data compared to the size of the internet. Indeed, any large queryable database or storage system goes to great pains to organize data to make it searchable. The rule of thumb is that if a storage system has $O(N)$ items of data, it can be searched over k keywords in only $O(k)$ operations and accessing approximately $O(k*r)$ bytes, where r is the record size per keyword. The technique for keyword search proposed in this article requires searching $O(N)$ items of data, a much larger term than $k*r$. As a result, it is fundamentally non-scalable as a search mechanism when $N \gg k*r$. Based on my interpretation of the results, there is no demonstration of scalability, since all of the proposed techniques rely on searching each file. Even if the search is carried out in parallel and even if some searches are carried out in the same oligo pool, it will become a massive cost to search all files and would be infeasible if not intractable to do on every query of a scaled-up system. For this reason, I do not share the author's enthusiasm for their approach. For the use case described, namely, keyword search, searching every file falls short of the efficiency of other methods.

I would like to have seen a better justification for querying a storage system in the proposed manner. Claims are made that this will be important in the future. Imagine a system with millions of files, each one needs to run the query and have a sensor detect if a match is present. This implies an extremely large number of searches with supporting sensors/hardware. Given the reported false positive rate, for millions of files, this would imply thousands of false matches requiring sequencing. Also, the cost of performing this query would be much worse than storing extra metadata that encodes where words occur and accessing a smaller amount of data with no false positives or false negatives.

A better explanation for why this kind of search should be done molecularly is needed.

(2) The proposed Non-Collision Grouping is not well explained, substantiated through a proper evaluation, or even needed, and is a major weakness of the paper. Despite the claims, it is actually

less efficient than simpler and straightforward alternatives. I examined and ran the implementation as provided on GitHub to make sure I understood the algorithm in its entirety.

There is a better approach that is likely more efficient and commonly used. After splitting the file into chunks along word boundaries, the remaining pieces can be split into m -bit sequences possibly with padding for boundary cases, call this ordered set of sequences M . (In this case, $m=16$ can be used to match the work in the manuscript, but it could be 8 or some other number. A small multiple of 8 is reasonable since that would preserve characters in their entirety.) Then, determine a new set made of only unique sequences in M , call this M' . The number of unique sequences is $|M'|$. And the minimum length of the required sequence is $\lceil \log(|M'|)/\log(b) \rceil$, where b is the number of symbols in the coding alphabet (4 for DNA, 3 for ternary, 2 for binary, etc). This is a known theoretical approach fundamental to coding theory. This should be considered as the baseline approach that should be compared. I believe this approach is simpler and equally, if not more, effective to the proposed approach.

I would suggest either proving beyond a shadow of a doubt that NCG helps, or, more likely, the paper would be better served by eliminating NCG and all associated claims in favor of a simpler, clearer discussion. You may not need to redo experiments, since you can simply define a transcoder that maps 16-bit sequences to 7-bit DNA sequences, and you are done, and you can justify it using the theoretical bounds you stated in the paper. However, if the NCG algorithm is left in the paper in its current form without more justification, I would not support publication because of the many unsubstantiated claims surrounding it.

Related to that, the argument that the NCG achieves a compressed result is unsubstantiated. What is really happening is that you are observing the fact that ASCII is a fixed length code and many symbols in that block code appear rarely because they are not often needed in English prose. (In other words, 8 bits per character per file is inefficient from an information theoretic perspective.) Also, relatively few combinations of letters make words as compared to all possible combinations. This is why you observe relatively few 16-bit sequences and why you are able to shrink 16 bits down to a few bases. It's not the NCG, rather, it's a consequence of your selection of text as the input set. Other inputs, like executables or images, would not have the same benefit. NCG is not really offering an advantage on its own. Rather, it's the relative sparsity of English vocabulary and repetition of common letters and words that allows your encoding to fit into 7 nt. You need to either show a convincing evaluation or re-write and revise to avoid unsubstantiated claims regarding the efficiency of NCG and resulting compression. This is a side effect of the inputs you selected and should not be conflated as a result of compression in the NCG algorithm, which it does not offer.

Finally, I suspect NCG is actually less efficient than the approach I describe above. This is because you encode each NCG group as an $N-1$ length sequence with an extra base as a pointer to pick the member of the group. Because you reserve a full base as the pointer, but many of your groups do not have 3 members, you are failing to use the full information density of the pointer most of the time. This implies the NCG is less efficient than it could be if it simply avoided grouping and packed all observed symbols into the $\lceil \log(|M'|)/\log(b) \rceil$ symbols.

For these two reasons, I have serious reservations about publishing the work. For the first one, it would require supplementing the work with a comparison of the complexity of this approach to existing approaches (digital ones) and acknowledging the inefficiency of the proposed methods, perhaps using a big-O comparison. You might say it's an unfair comparison, but the exact reason for algorithm or complexity analysis is to allow the comparison of approaches using radically different architectures and algorithms. It is exactly meant to help with this kind of scenario where computers may be implemented in radically different ways, digital vs molecular. Furthermore, the big-O notation helps understand the critical requirements and it reveals for this system that the approach is inefficient compared with known approaches. For the second concern about the NCG algorithm, either a convincing evaluation of NCG is needed or, more likely, the elimination of the NCG algorithm from the manuscript.

On the other hand, if we look beyond the specific application scenario and just consider the use of CRISPR plus a customized code to enable querying, there is novelty in this organization and it

demonstrates an interesting system for molecular querying. The most appropriate application and the best coding solution may not be in this work, but it lays a foundation for others to improve upon.

I have some other comments below.

+ Was each file isolated when searching? Or, are they always searched in an oligo pool containing a single file?

+ The error rate of 1.68% would be considered catastrophic and unusable in a conventional computer system. Typical error rates would be orders of magnitude lower, resulting from bugs or device failure. I suggest justifying this rate, pointing out that it's problematic for storage system design, and suggesting solutions.

+ Can you elaborate on the purpose of the reference strands? I think they are a record of the encoding choices that the NCG algorithm made. Why not store this record digitally to make the construction of a query easier? Also, why not use the same reference across many text files?

+ Line 183 - 187. Please elaborate on this calculation. Is it just estimated based on a sigmoid or is it measured?

+ When a query is carried out on a file, it cleaves the oligos that match but not the rest of the oligos in the file. Is it possible to perform independent queries so that an oligo pool can be used for more than one search?

With regard to reproducibility, the computational results appear to be reproducible. Although, it's not written with many comments. Many of the codes provided are inefficient. A significant concern is that no listing of required software packages is provided. It's recommended that the authors provide a standard requirements.txt file as is the custom for Python code bases. Please see <https://pip.pypa.io/en/stable/reference/requirements-file-format/>.

Reviewer #2:

Remarks to the Author:

The authors Zhang et al. investigate the viability of adopting a CRISPR-based method for keyword searches in DNA-based data storage systems. Through experiments, they find a near-linear correlation between the fluorescence intensity over a set duration and the number of queried keyword present in the file that is stored in the pool. This suggests a potential to estimate keyword counts in stored files without sequencing the entire pool. Compared to the in-tube tests, on-chip implementations produced noisier results but led to similar conclusions. Additionally, given that the method necessitates sequential text storage, the paper introduces a new coding scheme, NCG, which offers a more compact storage solution than directly storing the text information in DNA without compression. The simulations show that NCG is able to achieve a coding density slightly higher than 2 bits/nt in practical scenarios.

While the concept of "keyword-searching" in DNA-based storage is intriguing, I have the following concerns about the proposed approach.

- The proposed method is still not "sequencing-free", which is supposed to be the goal of the keyword-searching problem. To construct the query, one would need to first sequence part of the pool corresponding to the reference sequences, to know the NCG encoding dictionaries. However, if sequencing is needed anyhow, why not sequence the whole pool and search in the recovered text directly? The PCR amplification is of low cost compared to DNA synthesis, and one round of sequencing should provide enough coverage for all the encoded DNA sequences, at least for the data scale shown in the paper. So what are the key benefits of using the proposed method here? On the other hand, the drawback of the proposed approach is obvious, as one cannot use the well-developed compression methods for text anymore, resulting in a significant decrease in the storage density. It would be great if the authors can explain and analyze the trade-offs of the proposed method.

- Using CRISPR for text searching can be restrictive. As noted by the authors "the optimal range of length of a target being recognized by CRISPR reaction is 18-24 nt," most likely we are only able

to search one word with one query. I am wondering if the method can be adapted to enable the search for phrases and even sentences. Can it be achieved by a mixture of queries treated to the pool? Also, in current experiments, the keyword frequency is a bit low (max is 5). How will the system perform when this frequency is high (i.e., 100, 1000)? Will the fitted curve shown in Figure 3 (k) still be linear?

- Some arguments about the NCG coding are not correct. First in the SI, "Specifically, to reach Shannon capacity, we have: $n_2 \leq 4n_1 - 1$ ". If we are converting $8n_1$ bits into $n_2 - 1$ bases, the naïve relationship should be $2(n_2 - 1) \geq 8n_1$, leading to $n_2 \geq 4n_1 + 1$. And this will also affect the limit in SI equation (10), as the limit now should be smaller than 2. Furthermore, I do not think it is appropriate to analyze the asymptotic coding density based on the sigmoid assumption and empirical fitting of the parameters, and show the results in the main text. Figure 2 (a)-(f) in the main text is misleading and may not be the case in practice. If the authors want to analyze the coding density rigorously, they would need to develop probabilistic arguments to show the lower and upper bounds on the number of groups, given assumptions on the distributions of 0 and 1 at each position in the binary sequences.

- Besides the technical issues, the paragraphs describing the NCG coding can be improved. The authors show the performance of the coding scheme before explaining how the coding is done, which could confuse the readers. I think a more natural order should be first explaining the whole procedure clearly, then discussing the performance, otherwise, the readers will have no idea what role n_1 and n_2 are playing here and why the coding density would change with the data size.

- Some minor issues: There seem to be some typos in the manuscript. For example, in Figure 2 (i), after grouping the set of group ids is $\{640, 651, 716\}$, while for other parts of the figure it is $\{641, 650, 716\}$. Also, some flowcharts in the SI are not explained clearly. For example, in SI Figure 10, the authors want to show the procedure of finding the minimal set of non-collision groups such that the maximum group size is 3; however, the procedure shown in SI Figure 10 is misleading. I understand that the authors are trying to say they are using some greedy approach to obtain the groups, but I would recommend improving the readability of the flowcharts in the SI.

Response to Reviewer 1's Comments:

We appreciate Reviewer 1 for taking their valuable time to comment. Below, we have presented Reviewer 1's comments in italics and marked the corresponding responses in blue font.

Reviewer #1 (Remarks to the Author):

This is interdisciplinary work spanning computer system design and synthetic biology. And it's important to evaluate the novelty and value of the work from both perspectives. My expertise falls mostly on the computer systems side but I am also an expert in DNA-based storage and very familiar with the synthetic biology principles and protocols used in such systems.

Molecular search of data in a DNA storage system is a topic that has garnered attention recently and will likely continue to be of interest. Dense molecular storage systems, if scaled up to a cloud scale of exabytes and beyond, will eventually suffer an extreme IO bottleneck. The time it takes to access and sequence data will be very high limiting what can be sequenced to a small fraction of stored data at a time. If little is known about the data, then it would require prohibitive time and cost to first re-digitize it and then query it. Hence, efforts to directly search data molecularly are worth investigating. For these reasons, the paper would be of interest to a broad community of researchers across the computer system and synthetic biology spectrum. However, it's also important for the approaches explored in this area to build on the best approaches from each domain. I have significant concerns about aspects of this work from the computer systems perspective.

I have two major concerns with this work. (1) The search approach described is far too inefficient to be a useful search primitive in future scaled-out systems. It's only usable at a small scale where digital approaches work well. (2) The proposed NCG algorithm is not demonstrated to have novelty or to benefit the proposed system.

*(1) Assuming the high latency and low bandwidth sequencing platforms we have today, the problem of search exists in an extreme form for DNA-based storage. But, this problem already exists today. When a search query, using keywords, is sent to Google, Google doesn't look in every file on the internet to find the answer. Instead, it looks up meta-data that stores information about where words and terms appear and it consults a pre-computed answer for each term and then combines answers along with ranking to return a list of possible matches. This can be done rapidly, accessing very little data compared to the size of the internet. Indeed, any large queryable database or storage system goes to great pains to organize data to make it searchable. The rule of thumb is that if a storage system has $O(N)$ items of data, it can be searched over k keywords in only $O(k)$ operations and accessing approximately $O(k*r)$ bytes, where r is the record size per keyword. The technique for keyword search proposed in this article requires searching $O(N)$ items of data, a much larger term than $k*r$. As a result, it is fundamentally non-scalable as a search mechanism when $N \gg k*r$. Based on my interpretation of the results, there is no demonstration of scalability, since all of the proposed techniques rely on searching each file. Even if the search is carried out in parallel and even if some searches are carried out in the same oligo pool, it will become a massive cost to search all files and would be infeasible if not intractable to do on every query of a scaled-up system. For this reason, I do not share the author's enthusiasm for their approach. For the use case described, namely, keyword search, searching every file falls short of the efficiency of other methods.*

I would like to have seen a better justification for querying a storage system in the proposed manner. Claims are made that this will be important in the future. Imagine a system with millions of files, each one needs to run the query and have a sensor detect if a match is present. This implies an extremely large number of searches with supporting sensors/hardware. Given the reported false positive rate, for millions of files, this would imply thousands of false matches requiring sequencing. Also, the cost of performing this query would be much worse than storing extra metadata that encodes where words occur and accessing a smaller amount of data with no false positives or false negatives.

A better explanation for why this kind of search should be done molecularly is needed.

Response: We thank the reviewer for their positive comments on the importance and broad interest of our study. Additionally, we greatly appreciate the reviewer's insightful comments and concerns regarding the search efficiency and NCG algorithm of our proposed SEEKER method. These suggestions are valuable for further improving our manuscript.

We also appreciate the reviewer's expertise in interfacing computer systems and molecular biology. We share the same passion with the reviewer for developing a querying system to enable more convenient and reliable information extraction for applications in DNA data storage. This topic is so important that it warrants extensive study and perhaps requires many different approaches, particularly from the perspective of molecular biology, to be presented for reasonable comparison of these proposed methods in the future.

We understand the reviewer's criticism on the complexity of the searching scheme presented in this manuscript, and we fully agree that searches based on a computer system often rely on metadata with an inverted indexing data structure, which provides the location of a file that may carry the keyword. We believe that as the data size increases, it would be more appropriate and efficient to search metadata rather than the actual data. However, as for the general data search method proposed in the manuscript, SEEKER can certainly be adapted to search metadata with only mild modifications to the oligo data structure. We describe a scheme in **Supplementary Fig. 2a** inspired by recent similarity search work by Bee et al.¹.

Supplementary Fig. 2. Envisioning SEEKER in more advanced search settings. a, In a metadata search scheme, the features of a file, as the metadata, are pre-computed and encoded in a DNA strand along with the file index to assemble the payload. Each oligo is prefixed and suffixed with a set of PCR primer targeting sites to enable enrichment of the file. SEEKER can still be applied in this condition. When the query crRNA recognizes a matching feature, there will be fluorescence response in this enriched oligo pool, and no fluorescence will be observed if no matching feature is found. For the enriched pool with fluorescence signals, we can sequence the original pool using a small volume, or we can directly sequence the CRISPR reaction after SEEKER. **b**, The combination of SEEKER and hybridization-based search approaches enables rapid determination of the physical location of files containing the keyword and isolation of the file, which could make molecular searching in DNA database more convenient and efficient.

In this scheme, the features of a file, as the metadata, are pre-computed and encoded in a DNA strand along with the file index to assemble the payload. Each oligo is prefixed and suffixed with a set of PCR primer targeting sites to enable enrichment of the file. SEEKER can still be applied in this condition. When the query crRNA recognizes a matching feature, there will be fluorescence response in this enriched oligo pool, and no fluorescence will be observed if no matching feature is found. For the enriched pool with fluorescence signals, we can sequence the original pool using a small volume, or we can directly sequence the CRISPR reaction after SEEKER. We have demonstrated that amplified oligos, including those containing the keyword, will remain intact after the CRISPR reaction since no protospacer adjacent motifs (PAMs) were included in the sequence. Further elaboration can be found in the response to the reviewer's last minor concern. The decoding of sequencing reads will reveal which file contains the feature, and then we can read these particular files from the database.

An advantage of SEEKER under this search scheme is its ability to rapidly determine the physical location of the file containing the keyword. It is more probable that when handling large-scale DNA data storage, we tend to store oligo pools in different physical locations, perhaps in different chambers on a microfluidic device as demonstrated by Newman et al.². Imagine if we have no

prior knowledge of which chamber stores the keyword-containing files. We would have to enrich files from every chamber and subject them to searching, a process akin to the pre-amplification step in SEEKER. We believe that for every search method, there is an input capacity. Let us assume the amount of oligo stored in each chamber reaches this input capacity. If we have 1,000 chambers in parallel to screen, SEEKER allows us to digitally identify which chamber among the 1,000 contains the keyword through a fluorescence 0/1 state, with minimal sequencing reads required. If the dictionary is digitally stored, this search method can be completely sequence-free. We could even possibly avoid sequencing large amounts of data strands if there are no matches in the enriched files. However, with most existing methods, since there is no way to visibly determine which chamber contains the keyword, we must perform isolation and sequencing on every chamber. This significantly increases the workload, especially when only a few chambers contain the keyword.

However, we acknowledge that a weakness of SEEKER is the lack of physical isolation of strands carrying the keyword or the features being searched. Based on the above analysis of the strengths and weaknesses of SEEKER and conventional hybridization approaches, we believe the best way to address the molecular searching problem in DNA data storage is by combining these two methods (**Supplementary Fig. 2b**). SEEKER may be more useful as a pre-screening routine procedure to rapidly identify whether the keyword or feature exists in the enriched oligo pool. If so, the hybridization-based approach, potentially involving more complex operating procedures, can be implemented to isolate oligos from the candidate pools and read the metadata through sequencing. This way, both methods can complement each other to make molecular searching in DNA databases more convenient and efficient.

Next, although searching in metadata is more efficient, we would like to explain why we chose to demonstrate SEEKER-based searching in a way where the stored files were enriched and searched one by one. First, feature extraction involves computationally complex procedures, which are not the main focus of this study. Our expertise and research focus lie more in the molecular techniques used in DNA data storage applications rather than algorithmic design. Additionally, we believe that a simpler design, such as directly using the actual content of text files (each of which may contain hundreds of different words as the potential “keywords”), is more understandable to a broad audience including lay readers who are not experts in computer systems. Second, we believe it is more important to design and test many files, even though each file may be relatively smaller in size, rather than using a large-scale oligo pool to demonstrate the applicability of this system. On one hand, by testing numerous files, we can use a single query to conduct searches in different content with varying backgrounds, where hundreds of interfering keywords or features co-exist with the matching target, thereby testing the reliability of this system. On the other hand, we need to design repeated experiments with many files to validate that **1) files that are not supposed be accessed should not be amplified, which poses a challenge in the design of orthogonal PCR primer sets; and 2) content from unamplified files should not generate signals in the CRISPR reaction, otherwise catastrophic misidentification may occur.**

In this study, we designed 40 files each containing 200 words on average. We demonstrated that each query could generate correct 0/1 fluorescence responses with a ~1% error rate from a background of 8,000 potential targets. We believe this to be a significant achievement in feature extraction, one that has never been demonstrated before in DNA data search applications. As a rough estimate in response to the reviewer’s second minor concern, CRISPR-based searching

can recognize at least four continuous base mismatches in a 20-nt target, providing the precision to identify one keyword over a background of $4^{20} - 4^4 = 1.09 \times 10^{12}$ keywords.

We hope the above justification addresses the reviewer's concerns regarding why we adopted this search scheme to demonstrate the feasibility of SEEKER.

In this revised version, we have included a portion of the justification and analysis in the "System Overview" section of the manuscript, as well as in the Supplementary Information. Please refer to Lines 137-147 of the main text and Supplementary Fig. 2 in the Supplementary Information. In addition, we describe our proposed solution to address the keyword search problem in DNA data storage by combining SEEKER and conventional hybridization-based approaches in Lines 517-529 in the last paragraph of the Discussion section.

(2) The proposed Non-Collision Grouping is not well explained, substantiated through a proper evaluation, or even needed, and is a major weakness of the paper. Despite the claims, it is actually less efficient than simpler and straightforward alternatives. I examined and ran the implementation as provided on GitHub to make sure I understood the algorithm in its entirety.

There is a better approach that is likely more efficient and commonly used. After splitting the file into chunks along word boundaries, the remaining pieces can be split into m -bit sequences possibly with padding for boundary cases, call this ordered set of sequences M . (In this case, $m=16$ can be used to match the work in the manuscript, but it could be 8 or some other number. A small multiple of 8 is reasonable since that would preserve characters in their entirety.) Then, determine a new set made of only unique sequences in M , call this M' . The number of unique sequences is $|M'|$. And the minimum length of the required sequence is $\text{ceiling}(\log(|M'|)/\log(b))$, where b is the number of symbols in the coding alphabet (4 for DNA, 3 for ternary, 2 for binary, etc). This is a known theoretical approach fundamental to coding theory. This should be considered as the baseline approach that should be compared. I believe this approach is simpler and equally, if not more, effective to the proposed approach.

I would suggest either proving beyond a shadow of a doubt that NCG helps, or, more likely, the paper would be better served by eliminating NCG and all associated claims in favor of a simpler, clearer discussion. You may not need to redo experiments, since you can simply define a transcoder that maps 16-bit sequences to 7-bit DNA sequences, and you are done, and you can justify it using the theoretical bounds you stated in the paper. However, if the NCG algorithm is left in the paper in its current form without more justification, I would not support publication because of the many unsubstantiated claims surrounding it.

Related to that, the argument that the NCG achieves a compressed result is unsubstantiated. What is really happening is that you are observing the fact that ASCII is a fixed length code and many symbols in that block code appear rarely because they are not often needed in English prose. (In other words, 8 bits per character per file is inefficient from an information theoretic perspective.) Also, relatively few combinations of letters make words as compared to all possible combinations. This is why you observe relatively few 16-bit sequences and why you are able to shrink 16 bits down to a few bases. It's not the NCG, rather, it's a consequence of your selection of text as the input set. Other inputs, like executables or images, would not have the same benefit. NCG is not really offering an advantage on its own. Rather, it's the relative sparsity of English vocabulary and repetition of common letters and words that allows your encoding to fit into 7 nt.

You need to either show a convincing evaluation or re-write and revise to avoid unsubstantiated claims regarding the efficiency of NCG and resulting compression. This is a side effect of the inputs you selected and should not be conflated as a result of compression in the NCG algorithm, which it does not offer.

Finally, I suspect NCG is actually less efficient than the approach I describe above. This is because you encode each NCG group as an N-1 length sequence with an extra base as a pointer to pick the member of the group. Because you reserve a full base as the pointer, but many of your groups do not have 3 members, you are failing to use the full information density of the pointer most of the time. This implies the NCG is less efficient than it could be if it simply avoided grouping and packed all observed symbols into the ceiling($\log(|M'|)/\log(b)$) symbols.

Response: We appreciate the comments above. We understand the reviewer's concern about the potentially compromised information density with the NCG algorithm compared with the basic $\log|M'|/\log(b)$ method. It is true that in most cases, the overall coding potential with NCG is not superior to the $\log|M'|/\log(b)$ method. However, in the design of NCG coding algorithm, in addition to considering data compression ability, we placed greater emphasis on the base-economy with respect to the reference strands. In our scheme, the reference strands must be sequenced to obtain the query sequences before conducting any searches. Moreover, we expect by reducing the need to sequence reference strands, the overall cost of the writing-searching-reading process in DNA data storage can be lowered. We apologize for not making this point clearer in the manuscript.

In fact, the NCG algorithm is a variant of the $\log|M'|/\log(b)$ method suggested by the reviewer. If we still construct the dictionary and store it as reference strands in the oligo pool, the major difference between these two methods is the use of "Pointer" bases. For the NCG method, the "Pointer" base is necessary, as different "Pointer" bases refer to different data units in one group. Without the information of the "Pointer" base, there is no way to retrieve the data. With the $\log|M'|/\log(b)$ method, the "Pointer" base can be eliminated from the data strands if we pre-define a "Pointer" base or introduce a rule to map the "Pointer" base, which would apply to the retrieval of data units in all the groups.

When using the $\log|M'|/\log(b)$ method, more bases will be in the reference strands since the number of groups (M') can be greater or at least equal to what is generated by NCG algorithm. If we choose to store the dictionary along with the actual data in DNA, every time a portion of the oligo pool must be sequenced in order to determine the query sequence, allowing the searching to proceed. In response to the reviewer's third minor concern, for achieving the maximal overall coding density, one dictionary can only cover ~25–50 KB of data. This indicates that if we have a larger amount of data, multiple dictionaries are needed and all of them should be sequenced before molecular searches. With the sequencing of the dictionary becoming inevitable, we aim to minimize the cost of dictionary sequencing, making fewer reference bases undoubtedly more desirable. This becomes an advantage of using NCG instead of the $\log|M'|/\log(b)$ method.

To validate our method using experimental data, we randomly downloaded 100 MB of text data from Wikipedia and manually segmented it into 100 blocks with different block sizes from 5 KB to 1,000 KB. These 100 blocks were used for repeated experiments for one data size. We used both NCG and the basic $\log|M'|/\log(b)$ method to encode the data blocks and calculated the average number of reference bases per KB of data for different data sizes. As shown in **Supplementary Fig. 4a**, we found that there were always fewer reference bases for NCG, and the reduction of

reference bases was more significant for larger datasets. We then calculated the reduction of reference bases as a percentage and found that larger datasets led to a more significant reduction of reference bases for a data size up to 200 KB. The reduction reached its limit at ~15% for larger data sizes (**Supplementary Fig. 4b**). We also obtained the overall coding potentials for both methods and acknowledged that the basic $\log|M'|/\log(b)$ method typically had a higher overall coding potential, except in some cases where the number of groups for the basic method exceeded the encoding volume ($V = 3^{n_2}$, where n_2 is the number of bases in one data unit) and had to include one more base in the data unit, while the number of groups for NCG was still within the range (**Supplementary Fig. 4c**).

Supplementary Fig. 4. Comparison of NCG and the basic $\log|M'|/\log(b)$ method. **a**, The average number of reference bases per KB of data with NCG and the basic $\log|M'|/\log(b)$ method for different data sizes. **b**, The reduction of reference bases as a percentage for NCG compared with the $\log|M'|/\log(b)$ method. **c**, The calculated coding potential with both methods for different data sizes. For each block size, 100 different blocks were tested as experimental replicates. **d**, The overall cost per MB of data as a function of the ratio of oligos being sequenced in the entire oligo pool. The intersection where the costs for NCG and the basic $\log|M'|/\log(b)$ method were equal is marked. **e**, The number of reference bases per KB of data for both methods when the data was in non-text format. **f**, The coding potential for each 5 KB block of non-text data for both methods.

However, we are more interested in assessing the effectiveness of these two approaches in terms of the cost of oligo synthesis and sequencing. We aimed to investigate whether, in cases where the sequencing of reference bases becomes inevitable, a reduced need for reference base sequencing will lead to an overall reduction in the cost of DNA data storage. With advancements in microchip technology, the cost of synthesis per nucleotide can potentially be reduced to \$0.00001.³ According to online data (<https://ourworldindata.org/grapher/cost-per-gigabase-dna-sequencing>) the cost of sequencing per gigabase was \$5.04 in 2021. Based on these statistics, we can roughly estimate the overall cost of writing and reading in DNA using both the NCG and $\log|M'|/\log(b)$ methods. We investigated a case where 1 MB of data is stored in one dictionary, resulting in a reduction of reference bases to ~15% when using the NCG method. A previous study reported that information with a density of 215 PB/g can be recovered through sequencing⁴. Assuming a yield of 10 ng for the oligo pool and that the product from first PCR can be used at least once as the substrate in a new PCR without significant quality reduction, and since each

PCR requires only 0.00465 ng to access 1 MB of data, the original oligo pool can be used $2,150^2$ times without the need to synthesize a new pool. We have summarized the costs of synthesis and sequencing for these two methods in **Supplementary Table 2**.

Method	Coding Potential (bits/nt)	Synthesis Cost per MB	Average Reference Base per MB	Reference Sequencing Cost per MB ($2,150^2$ times)	Data base per unit	Data Sequencing Cost per MB ($2,150^2$ times)
NCG	2.0	\$40	25,180	\$586.74	8	\$93189.6
$\log M' /\log(b)$	2.3	\$34.78	29,318	\$683.0	7	\$81540.9

Supplementary Table 2. Statistics of the cost of synthesizing and sequencing data stored in DNA with the NCG method and the basic $\log|M'|/\log(b)$ method.

Since the aim of searching is to find the content-of-interest, which is expected to take up only a portion of the entire dataset, we define the ratio of oligos encoding the content-of-interest as the x-variable. We assume only this portion of data bases is sequenced. Using the statistics in **Supplementary Table 2**, we calculated the overall cost per MB of data as a function of x (**Supplementary Fig. 4d**). The point of intersection between the lines represents the point at which the costs for these two methods are equal. We found that when the ratio of oligos encoding the content-of-interest was below 0.78%, the NCG method had an overall lower cost compared to the $\log|M'|/\log(b)$ method. In practical conditions, it is common for only a very small fraction of oligos in the entire oligo pool to be sequenced. Taking Wikipedia as an example, there are only 11 words with a frequency ratio higher than the threshold of 0.78% (https://www.thingsmadethinkable.com/item/words_on_wikipedia.php). These words mostly consist of determiners such as 'the', prepositions such as 'of', 'in', 'to', 'on', and 'for', and conjunctions such as 'and' and 'as', which are not suitable for use as keywords. Therefore, assuming that the occurrence of words is evenly distributed throughout the Wikipedia corpus, the ratio of files containing a particular keyword will certainly be lower than 0.78%. This suggests that NCG may be a better choice for achieving a lower cost in the complete writing-reading-searching process in DNA data storage.

Moreover, to respond to the reviewer's concern that NCG solely depends on the features of the English vocabulary to achieve compression, we would like to demonstrate several more examples to extend the application of NCG from texts to pictures in JPG format and animation in GIF format, as well as some randomly generated binary codes. The names of the encoded files are listed in **Supplementary Table 3**:

Shorthand	File name	Size
JPG-1	322px-Mona Lisa.jpg	43.9 KB
JPG-2	380px-sunflowers.jpg	56.5 KB
GIF-1	DNA_orbit_animated.gif	660 KB
GIF-2	Newtons_cradle_animation.gif	301 kB
Random-1	Random binary codes-I	50 KB
Random-2	Random binary codes-II	50 KB

Supplementary Table 3. Name and size of files in non-text format to be encoded with NCG and the basic $\log|M'|/\log(b)$ method.

To encode all the files, we applied an 8-bit grouping interval, considering every 5 KB of data as a segment or block for encoding. We calculated the number of reference bases per KB and the overall coding potential, which can be found in **Supplementary Figs. 4e-f**. As shown in **Supplementary Fig. 4e**, the number of reference bases in the NCG algorithm reduced to about 60% of what was generated through the basic $\log|M|/\log(b)$ algorithm. This means that if it becomes inevitable to recover the dictionary for a file, the bases that need to be sequenced under the basic method will be 1.6 times that of those under the NCG method, thus increasing the sequencing price and time cost. As shown in **Supplementary Fig. 4f**, with certain block sizes the coding potential for NCG was even slightly higher than that of the basic method, although one more “pointer” base was consumed. This is because the reduction in the number of reference bases was more significant than the reduction in the number of data bases when a proper grouping interval was applied. This phenomenon is independent of the data format and serves as evidence that NCG benefited not solely from the sparsity of the English vocabulary and the repetition of common letters and words, but also from the “non-collision grouping” mechanism.

In the revised version, we have included the above justification in Section 2.4 of the Supplementary Information. Based on the results obtained, we briefly discussed the benefits of using NCG in Lines 166-176 of the main text. We hope this justification has made our points clearer. As per the reviewer’s request, we have removed most parts of the discussion about the NCG algorithm from the main text and only kept the explanation of the oligo data structure to maintain readability of the manuscript.

For these two reasons, I have serious reservations about publishing the work. For the first one, it would require supplementing the work with a comparison of the complexity of this approach to existing approaches (digital ones) and acknowledging the inefficiency of the proposed methods, perhaps using a big-O comparison. You might say it’s an unfair comparison, but the exact reason for algorithm or complexity analysis is to allow the comparison of approaches using radically different architectures and algorithms. It is exactly meant to help with this kind of scenario where computers may be implemented in radically different ways, digital vs molecular. Furthermore, the big-O notation helps understand the critical requirements and it reveals for this system that the approach is inefficient compared with known approaches. For the second concern about the NCG algorithm, either a convincing evaluation of NCG is needed or, more likely, the elimination of the NCG algorithm from the manuscript.

Response: Thank you for these comments. For the first concern, as suggested by the reviewer, we listed the major existing algorithms used in a search engine and made a comparison regarding the complexity in Big-O notation. The list includes the algorithm presented in this work, which we believe can be categorized as a linear search. Please refer to the list in **Supplementary Table 1**:

Name of Algorithm	Complexity	Notes
Inverted Indexing	O(1) for exact term lookup O(log(N)) for phrase queries	N is the number of documents containing the keyword
TF-IDF (Term Frequency-Inverse Document Frequency)	O(N × M)	N is the number of documents and M is the number of unique keywords in a query

PageRank	$O(V^3)$	V is the number of pages in the web graph
Cosine Similarity	$O(N \times M)$	N is the number of documents and M is the number of unique keywords in a query
B-Trees	$O(\log(N))$	N is the number of keys in the tree
BM25 (Best Matching 25)	$O(N \times M)$	N is the number of documents and M is the number of unique keywords in a query
SVD (Singular Value Decomposition)	$O(N \times M \times 2)$	N is the number of documents and M is the number of unique keywords in a query
Linear search (as demonstrated in this work)	$O(N \times M)$	N is the number of documents and M is the number of unique keywords in a query

Supplementary Table 1. List of major existing search algorithms employed in search engines and their complexity.

Although the method demonstrated in this work is more approximate to a simple linear search, it has quantitative search capability by counting keyword frequency, similar to TF-IDF. These two techniques share the same complexity. Moreover, our method is comparable in complexity to some widely used search algorithms that have the ability to return the relevance of documents to a query, such as Cosine Similarity, BM25, and SVD. Therefore, this method should be considered a valid technique. For the inverted indexing technique with a “metadata + file ID” configuration, our concern is that to conduct quantitative searching based on keyword frequency, a significant amount of additional memory will be consumed to store the pointers, the metadata, and their relevance to the file. However, we acknowledge that data structures like inverted indexes can dramatically improve search efficiency. Thus, we proposed a supplemental SEEKER-based search scheme with oligos designed for inverted indexes, which is shown in **Supplementary Fig. 2**. The complexity comparison of algorithms used in search systems can be found in Section 1.2 of the Supplementary Information.

For the second concern, we followed the reviewer’s suggestion to remove the performance analysis of NCG from the main text. However, to maintain fluency in reading for potential readers still interested in the coding scheme, we have included the detailed explanation of the algorithm and some modified performance analysis in the Supplementary Information. While not elaborating on the algorithm in this section, we believe it is necessary to mention how the reference strands and data strands were structured, as well as the range of coding potentials under this data structure, to make the following content more intelligible to readers. Please refer to the modified Fig. 2 of the main text.

On the other hand, if we look beyond the specific application scenario and just consider the use of CRISPR plus a customized code to enable querying, there is novelty in this organization and it demonstrates an interesting system for molecular querying. The most appropriate application and

the best coding solution may not be in this work, but it lays a foundation for others to improve upon.

Response: Thank you for your positive comments!

I have some other comments below.

+ *Was each file isolated when searching? Or, are they always searched in an oligo pool containing a single file?*

Response: Thank you for your question. There is no need to isolate the file when searching. The oligo pool we used for searching with the four queries (**Fig. 3i**) actually contained 40 files. Each file was PCR amplified before searching. After amplification, the majority of the oligos in the oligo pool are from the amplified file. Therefore, the searching can proceed without interference from other files. The ability to perform keyword searching in a particular file over a complex mixture of files in an oligo pool originated from the idea presented by Organick et al.⁵ with the help of careful PCR primer design (**Supplementary Fig. 5**). It is also a key feature of this SEEKER search system.

+ *The error rate of 1.68% would be considered catastrophic and unusable in a conventional computer system. Typical error rates would be orders of magnitude lower, resulting from bugs or device failure. I suggest justifying this rate, pointing out that it's problematic for storage system design, and suggesting solutions.*

Response: Thank you for your concern regarding the error rate presented in our study. We agree with the reviewer's opinion that error rate is a vital aspect of a searching system, and an error rate of around 1–2% is too high for a searching system to be considered practical. However, we believe that from the perspective of molecular biology, CRISPR, or some other enzyme-based methods, offer better specificity in nucleic acid sequence identification compared to basic hybridization-based approaches. This implies that the high error rate can only be resolved through sequence design, which requires more stringent criteria to ensure the orthogonality of sequences encoding unique keywords or features.

We designed experiments to compare the performance of the conventional DNA hybridization method and CRISPR-based methods such as SEEKER in recognizing mismatched sequences. The query is a 20-nt ssDNA corresponding to the keyword 'storage' as used in our study. In a simplified experimental design, we used this query to recognize different targets, including the fully complementary one and targets with continuous four-base mismatches. To avoid massive continuous matches between the query and a mismatched target, which could more possibly generate non-specific signals for both the basic DNA hybridization and the SEEKER method, the mismatched bases were designed in the middle of the target sequence. The sequences of the query and target DNA, whether matched or mismatched, are listed in **Supplementary Table 8** (red domains indicate the mismatched bases):

Name	Sequence (5'-3')
crRNA-storage_1	UAAUUUCUACUCUUGUAGAUUAGCGUCACGCGUCGCUCGUC

Fully matched DNA target	GACGAGCGACGCGTGACGCTA
Mutation_1	GACGACGCTCGCGTGACGCTA
Mutation_2	GACGAGCGAGCGCTGACGCTA
Mutation_3	GACGAGCGACGCGACTGGCTA

Supplementary Table 8. Oligo sequences for experiment comparing the specificity between SEEKER and the conventional hybridization approach (red bases refer to mismatches).

For the SEEKER method, the specificity was measured through real-time monitoring of fluorescence resulting from CRISPR-based trans-cleavage of ssDNA reporters, using the same mechanism introduced in this study for keyword identification. In the basic hybridization method, we mixed the query with the targets, incubated the reaction at room temperature for 1 h, and performed native gel electrophoresis to measure the extent of binding between the query and targets. In both approaches, the concentration of the query and all target species was fixed to 500 nM with a 1:1 ratio.

Supplementary Fig. 19. Comparison of specificity in sequence identification between SEEKER and the hybridization-based approach. **a**, The normalized trans-cleavage rate for mismatched and fully matched targets in SEEKER. The average trans-cleavage rate for three experimental repeats for the fully matched target was normalized to 1.0. The results indicate that all mismatched targets had a trans-cleavage rate significantly lower than the fully matched one, with the highest among them reaching a value of 0.2. **b**, Gel electrophoresis showing the binding affinity between the query and targets. The band indicating hybridization between mutated target-2 and the query was even darker than that for the reaction between the fully matched target and the query. **c**, Binding affinity revealed by measuring the band intensity. The band intensity for mutation-2 was almost four times the intensity for the fully matched target, suggesting a possibly much stronger binding affinity between the query and a mismatched target. The error bars represent the means \pm standard deviation (s.d.) from three replicates. The list of oligo sequences used in this experiment can be found in Supplementary Table 8.

For the SEEKER method, we calculated the trans-cleavage rate for every target, which was defined as the average fluorescence enhancement per minute in the initial stage of the reaction. We normalized the trans-cleavage rates in which the trans-cleavage rate for the fully matched target was the maximum (1.0). The results are shown in **Supplementary Fig. 19a**. From the results, it is clear that all mismatched targets had a significantly lower trans-cleavage rate than the fully matched one, with the highest among them reaching a value of 0.2. For the basic hybridization method, the gel image is shown in **Supplementary Fig. 19b**, and we observed that

the band indicating hybridization between mutated target-2 and the query was even darker than that for the reaction between the fully matched target and the query. To better characterize this, we measured the intensity of the hybridization bands using ImageJ and found that the band intensity for mutation-2 was almost four times the intensity for the fully matched target (**Supplementary Fig. 19c**), suggesting a possibly much stronger interaction between the query and a mismatched target.

Apparently, catastrophic misidentification in keywords or features is more likely to happen when applying the basic hybridization method. This problem will become especially serious if the system is designed to have low tolerance in recognizing extremely similar sequences but should only recognize the exact one, given two words common in some letters are encoded in sequences with overlaps. For instance, 'father' and 'mother' share 'ther' at the end. However, we believe enzyme-based methods like SEEKER will alleviate this problem by improving the specificity in sequence recognition. From this perspective, SEEKER may also make sequence design less challenging, as the requirement to find "extremely" different sequences becomes less strict, allowing for easier, more logical, and possibly more computable mapping of features to sequences.

*In this revised version, we have included these results in **Supplementary Fig. 19** and discussed potential solutions to reduce the error rate in **Lines 303-309** of the main text.*

+ *Can you elaborate on the purpose of the reference strands? I think they are a record of the encoding choices that the NCG algorithm made. Why not store this record digitally to make the construction of a query easier? Also, why not use the same reference across many text files?*

Response: We appreciate the reviewer's question about the purpose of reference strands and their storage. As the reviewer mentioned, the reference strands store the dictionary we used to map the 7-nt sequence unit in the data strands to the binary code of the text, and vice versa. In this way, the sequence of the query can be determined by reading the reference strands, which occupy a very small proportion of the entire oligo pool, allowing searching to proceed.

We abandoned the idea of storing the reference digitally because our aim is to construct and demonstrate a digital-free storage system that is entirely based on DNA storage. In this study, we only presented one dictionary, which covered around 40 KB of data. If we store the dictionary digitally in its current form, which contains 717 groups each with 3 elements with a maximum of 16-bit combinations, the total number of bits required to store the reference would be $717 \times 16 \times 2 = 22,944$ bits = 2.868 KB. Although the dictionary takes up about 5% of the storage space, if we scale up the data volume to the GB or TB level, the storage space required for the reference may also be increased to the GB or TB level. Considering the data to be stored in DNA are mostly archived data, which are not frequently accessed, the dictionary storing the information to decode these data should also be rarely accessed. Therefore, we believe there is no need to use external digital space to store such a large dictionary, and rather we can synthesize the dictionary along with the actual data and store them in DNA.

As to using the same dictionary across many text files, it is indeed helpful to reduce the size of dictionary and make it reasonable to be stored digitally. However, based on our selection of the parameters for the NCG algorithm, effective data compression of text data into DNA bases can only be achieved when the number of groups does not exceed the encoding volume ($V = 3^{n_2-1}$), which is 729 in this work. This implies that the size of the dictionary should be restricted and only

cover a certain amount of data in order to achieve the maximum coding potential. Considering the ideal grouping interval ($G = 8n_1$), which is 16 as demonstrated in our study, two ASCII symbols are encoded in one unit, resulting in $256 \times 256 = 65,536$ possible combinations in a unit. If all these combinations are taken into consideration, there will be $65,536/2 = 32,768$ non-collision groups if we ideally group each code with its complement code (e.g., “1000000000000000” and “0111111111111111”). The base-3 form of 32,768 contains 10 digits, which refers to 10+1 bases for one data unit in the data strands. If using the suggested $\log|M'|/\log(b)$ method, there will be 10 bases in each data unit, resulting in a coding potential roughly estimated to be 1.6 bit/nt, significantly smaller than the coding potential with fewer bases in one data unit plus the reference bases, and it would theoretically not be possible to achieve data compression. By definition, the data unit can contain no more than 7 bases to ensure the coding potential is theoretically higher than 2 bits/nt. Even if we utilize the feature in the English language and only consider commonly occurring letter combinations, the number of groups will easily exceed 729 if we intend to store around 50 KB or more data within one dictionary, which compromises the coding potential. In summary, although using one dictionary to encode all files will greatly reduce the number of reference bases and avoid the need to sequence dictionaries before reading the file, we are concerned that more cost will be added to oligo synthesis, and therefore it may not be as economical as expected.

+ Line 183 - 187. Please elaborate on this calculation. Is it just estimated based on a sigmoid or is it measured?

Response: We appreciate the reviewer’s question about our estimation of the limit of the number of groups with different data sizes. We are aware that the estimation based on sigmoid fitting was not appropriate and might be inaccurate if large data sizes are investigated. We therefore estimated the limit of coding potential from both a theoretical perspective, based on the consideration of extreme conditions of data composition, and a practical perspective through a probabilistic model using real input data.

The foundation of NCG in data compression mainly relies on the elimination of repeated and “collided” data units by assigning them to one group, and the coding density is mainly influenced by the number of groups that further influences both the number of bases in the reference strands and data strands. Thus, the best-case scenario is when the data only contain three unique units, and these units happen to be “non-collided” and are therefore assigned to one group. The worst-case scenario is when the data contain no repeated units and no “non-collision” units are found, and thus each group only includes one data unit.

In the worst-case scenario, suppose we have N symbols, making up $8N$ bits of data in total. Given the grouping interval $G = 8n_1$, if all the data units are unique, the number of groups should be:

$$N_G = \frac{8N}{8n_1} = \frac{N}{n_1} \quad (1)$$

Then, the number of bases in the reference strands should be the same as the data size:

$$B_{rp} = 8n_1 \cdot \frac{N}{n_1} = 8N \quad (2)$$

The number of bases in the data strands will be:

$$B_{dp} = \frac{8N}{8n_1} \cdot (\log_3 N_G + 1) = \frac{N}{n_1} \left(\log_3 \frac{N}{n_1} + 1 \right) \quad (3)$$

As this analysis is for theoretical estimation of the upper and lower limit of the coding potential, we assume the number of data bases to encode one data unit changed continuously rather than by one base if the threshold is reached, and the encoding volume n_2 is no longer considered in the following analysis.

Considering each binary code has its non-collided “complement code” (i.e., “0000100” and “11111011”), the ASCII code table containing 256 symbols can be divided into two parts, the original codes and the complement codes, in which each original code can pair with its complement code and be assigned to one group, but the original codes can only be assigned to different groups. In this case, when the grouping interval is 8 ($n_1 = 1$), which means one data unit contains one ASCII code, there are $2^7 = 128$ original codes and 2^7 groups in maximum. When the data size is below or equal to 2^7 bytes, in the worst-case scenario we assume the data is all composed of original codes. Then, the coding potential when ($N \leq n_1 \cdot 2^7 = 2^7$) can be calculated as:

$$P = \frac{8N}{8N + N(\log_3 N + 1)} \quad (4)$$

When the data size is larger than 2^7 bytes, which refers to $N > 2^7$, the coding potential is:

$$P = \frac{8N}{8 \times 2^7 + N(\log_3 2^7 + 1)} \quad (5)$$

Similarly, when the grouping interval is 16 ($n_1 = 2$), there are 2^{15} original codes. When $N \leq n_1 \cdot 2^{15} = 2^{16}$, the representation of the coding potential is:

$$P = \frac{8N}{8N + \frac{N}{2} \left(\log_3 \frac{N}{2} + 1 \right)} \quad (6)$$

When $N > 2^{16}$, the coding potential is:

$$P = \frac{8N}{8 \times 2^{16} + \frac{N}{2} (\log_3 2^{15} + 1)} \quad (7)$$

We can generalize the representation of the coding potential with any n_1 as:

$$P = \begin{cases} \frac{8N}{8N + \frac{N}{n_1} \left(\log_3 \frac{N}{n_1} + 1 \right)}, & 0 < N \leq n_1 \cdot 2^{8n_1-1} \\ \frac{8N}{8n_1 \cdot 2^{8n_1-1} + \frac{N}{n_1} (\log_3 2^{8n_1-1} + 1)}, & N > n_1 \cdot 2^{8n_1-1} \end{cases} \quad (8)$$

In the best-case scenario, we assume the data are ideally composed of repeats of three “non-collided” data units, which can be exactly assigned to one group, resulting in only $8n_1$ bases in the reference strand. Then, the coding potential can be represented as:

$$P = \frac{8N}{8n_1 + \frac{2N}{n_1}} \quad (9)$$

Based on the above analysis, we can then determine the lower and upper bounds of the coding potential as a function of symbol number N and grouping interval n_1 . With data size as the X-axis, the upper and lower bounds of the coding potential under different n_1 are plotted in **Supplementary Fig. 3a**, where the value of n_1 is marked in each subfigure.

Supplementary Fig. 3. Theoretical and practical estimation of the coding potential for NCG. **a**, Theoretical upper (red curves) and lower (blue curves) bounds of the coding potential under different n_1 based on consideration of extreme conditions of data composition. The value of n_1 is marked in each

subfigure. **b**, The lower bound of the coding potential when $n_1 \leq 8$ with data sizes up to 1,000 MB. **c**, The actual coding potential obtained with real text data from Wikipedia when $n_1 = 1, 2, 3$. Each data size was tested using 10 different blocks as experimental replicates. **d**, The normal distribution of the number of groups when $n_1 = 2$. Block sizes are marked in each subfigure. For each block size, 100 different blocks were tested as experimental replicates. **e**, The measured results of the number of groups from the 100 replicates with different block sizes. **f**, The calculated upper and lower limit of the number of groups with every block size based on a probabilistic estimation of the normal distribution. **g**, The upper and lower limit of coding potential for every tested block size based on the estimated upper and lower limit of the number of groups.

As the best-case scenario reflects an ideal condition that may rarely happen with real-world data, we speculated that the worst-case scenario may be more approximate to actual coding potentials. Through comparison of the lower theoretical bounds, we were able to determine the best grouping interval and the range of data block sizes to be applied in NCG coding. With the current stage of DNA data storage mostly involving data sizes at the MB level, we calculated the lower bound of coding potential under different n_1 with data sizes up to 1,000 MB (**Supplementary Fig. 3b**). We found in the cases of $n_1 = 1$ and $n_1 = 2$, the lower bound rapidly grew when the data size was less than 1 MB. When $n_1 = 2$, the lower bound was the highest across the studied data size range. Finally, when $n_1 = 3$ the lower bound gradually increased but was still slightly lower than the case of $n_1 = 2$ when the data size reached 1,000 MB. For higher n_1 values, the lower bounds remained significantly lower. This result suggested that $n_1 = 2$ might be a better choice as the grouping interval for NCG for text data encoding.

Then, we aimed to provide measured upper and lower bounds based on a probabilistic model using real input data, which can better reflect the actual conditions of text data encoding. To demonstrate this, we downloaded ~100 MB of text data from Wikipedia as the input. The original data were divided into multiple blocks with a certain size, and different block sizes were studied to reveal the relationship between data size and the number of groups under the NCG algorithm. For each block size, 100 blocks were computed for experimental repeats, and we expect the number of groups in these 100 experiments to follow a normal distribution. This allows us to estimate the upper and lower limits from a probabilistic perspective. The block sizes chosen were 5 KB, 10 KB, 25 KB, 50 KB, 100 KB, 250 KB, 500 KB, and 1,000 KB.

Before moving on to the measurement of the number of groups, we aimed to confirm the conclusion drawn from the theoretical analysis that $n_1 = 2$ should be applied to NCG coding. We analyzed the coding potential when the n_1 value ranged from 1 to 3 using real text data from Wikipedia, testing 10 different blocks as replicates for each size. We confirmed that $n_1 = 2$ had a better actual coding potential with data sizes below 50 KB (**Supplementary Fig. 3c**). When the data size further increased, although the coding potential for $n_1 = 3$ might increase, the number of groups would increase even more dramatically, resulting in much longer computation times and greatly reducing the efficiency. Therefore, we believe it is reasonable to use $n_1 = 2$ as the grouping interval.

With the grouping interval n_1 determined as 2, the distributions of the number of groups are shown in **Supplementary Fig. 3d**, with block sizes marked in each subfigure. For most block sizes, the number of groups in the 100 experiments followed a normal distribution. The measured results of the number of groups are also shown in **Supplementary Fig. 3e**. We observed an increase in the average number of groups as the block size increased, with what appeared to be a linear

relationship with the logarithm of the block size. However, there was substantial variation in the number of groups when the block size became large, which led to an even fewer number of groups for a larger block size in some cases.

We define the upper and lower limit as when the significance level reached 0.01, where the probability of observing a number of groups exceeding the defined upper limit or below the defined lower limit is 0.01. To calculate the upper and lower limit, we used the Z-score method:

$$Z = \frac{x - \mu}{\sigma} \quad (10)$$

where x is the observed value, μ is the sample mean, and σ is the standard deviation. From the Z-score table, the Z-value should be 2.33 for a significance level of 0.01. Thus, the upper and lower limit are calculated as:

$$x_L = \mu \pm 2.33\sigma \quad (11)$$

The upper and lower limits of the number of groups with every block size tested are shown in **Supplementary Fig. 3f**. The upper and lower limits rapidly increased when the data size was smaller than 100 KB. When the data size further increased, the lower limit reached a stable value at around 1,000, while the upper limit still increased but at a slower rate. We anticipate that the upper limit would soon reach its threshold if the data size were further expanded, due to the fact that the number of frequent character combinations in the English text is limited. We also calculated the practical upper and lower limit of coding potential for every block size we tested based on the estimated upper and lower limit of the number of groups. The results are shown in **Supplementary Fig. 3g**. It should be noted that in practice, when the number of groups exceeds $3^6 = 729$, an additional base must be added to each data unit of the data strands, which may compromise the overall coding potential. Consistent with our speculation, the optimal upper limits of the coding potential were obtained with data sizes of 25–50 KB, which corresponded to a measured number of groups close to 729 (**Supplementary Fig. 3e**). As the data sizes further enlarged, the coding potential dropped back to below 2 bits/nt but kept approaching 2 bits/nt. We concluded that in this coding scheme, to achieve a higher coding potential the data size encoded with one dictionary should be limited to 25–50 KB, and more dictionaries are recommended if encoding data with larger sizes than the optimal range.

In this revised version, we have included the above analysis in Section 2.3 of the Supplementary Information.

+ *When a query is carried out on a file, it cleaves the oligos that match but not the rest of the oligos in the file. Is it possible to perform independent queries so that an oligo pool can be used for more than one search?*

Response: We appreciate the reviewer's comment on the reusability of the SEEKER system. The answer is yes, it is true that the CRISPR-Cas12a system will cleave the double-stranded DNA targets, but effective cleavage only occurs in sequences with a PAM motif⁶. We did not include any PAMs in our sequence design of the data strands in the SEEKER system. Therefore, the targets were not degraded when performing SEEKER, and the reaction after one search can definitively be applied in the next search.

However, to simplify the experimental procedure of the reusable SEEKER, the cleaved ssDNA reporters are not removed, which means the fluorescence generated from the last reaction will be brought to the next one. At the same time, the volume of the last reaction introduced into the next reaction directly influences the fluorescence enhancement, as the last reaction carries the amplicons. If we reuse the amplicon multiple times, the quantity of the targets will exponentially decay. Therefore, it is of great importance to study the impact of the amplicon sample volume on the effectiveness of reusable SEEKER.

Another concern is that the reaction from the last SEEKER search contains the previous query crRNA, which could cause interference and result in misidentification if it remains in the next reaction. To eliminate such interference, we processed the last reaction with RNase I_f, which degrades RNA molecules without affecting the amplicon DNA. The RNase I_f processing protocol involves incubation at 37°C for 15 min followed by heat inactivation at 85°C for 30 min.

In the experiment, we selectively amplified File #33 from the oligo pool used for the molecular search in this work. First, we used the query 'comput' to search for content in File #33, and the fluorescence was expected to saturate. Next, we treated the sample with RNase I_f to remove the previous query crRNA. For the next round of the SEEKER search, we used the query 'database', which was present in File #33, and the query 'storage', which was absent. Different sample volumes ranging from 1.5 µL to 6 µL were added to the SEEKER reaction to investigate the impact of the amplicon sample volume on the reaction performance. Depending on the performance, more rounds of reusable SEEKER can be conducted. A schematic illustration of the experimental design is shown in **Supplementary Fig. 22a**.

As shown in **Supplementary Fig. 22b-e**, we learned that the fluorescence kinetics for reusable SEEKER were different from those of the first-round SEEKER. The fluorescence intensity would initially decay, possibly due to the introduction of RNase I_f or its solvent components. However, the distinction between reactions with a matched query and an unmatched query was still evident. In reactions with a matched query, the fluorescence intensity started to increase at about 30 min (**Supplementary Fig. 22b**), whereas it continued to decay with an unmatched query, with only a slight increase after 120 min of reaction (**Supplementary Fig. 22c**). For the third round of SEEKER, there was no obvious fluorescence enhancement in the first 120 min of reaction with both queries (**Supplementary Fig. 22d and e**). However, in the reaction with matched queries, a measurable fluorescence increase was observed after 120 min (**Supplementary Fig. 22d**), while no apparent increase was found with an unmatched query (**Supplementary Fig. 22e**). Based on the results, the reusable SEEKER can be implemented at least three times, and we speculate the fluorescence enhancement with matched queries would experience a more significant decay and become negligible if more rounds were performed. In addition, we confirmed that the fluorescence baseline would be higher if more volumes from the last reaction were applied, but the fluorescence slope was not apparently enhanced with additional volumes from the last reaction.

Supplementary Fig. 22. Reusable SEEKER. **a**, The scheme of experimental design of reusable SEEKER. In the experiment, we selectively amplified File #33 from the oligo pool we used for the molecular search in this work. First, we used the query 'comput' to search for content in File #33, and the fluorescence was expected to saturate. Next, we treated the sample with RNase I at 37°C for 15 min followed by heat inactivation at 85°C for 30 min to remove the previous query crRNA. For the next round of the SEEKER search, we used the query 'database', which was present in File #33, and the query 'storage', which was absent. Different sample volumes ranging from 1.5 μL to 6 μL were added to the SEEKER reaction to investigate the impact of the amplicon sample volume on the reaction performance. Depending on the performance, more rounds of reusable SEEKER can be conducted. **b**, Real-time fluorescence of second-round SEEKER with a matched query. **c**, Real-time fluorescence of second-round SEEKER with an unmatched query. In Figs. b-c, the fluorescence intensity started to increase at about 30 min with the matched query, but continued to decay with an unmatched query, with only a slight increase after 120 min of reaction. **d**, Real-time fluorescence of third-round SEEKER with a matched query. **e**, Real-time fluorescence of third-round SEEKER with an unmatched query. In Figs. d-e, there was no obvious fluorescence enhancement in the first 120 min of reaction with both queries. However, in the reaction with matched queries, a measurable fluorescence increase was observed after 120 min, while no apparent increase was found with an unmatched query. **f**, Slope of fluorescence for the second-round SEEKER with a different sample volume from the last reaction. The fluorescence enhancement in the first 90 min was considered. **g**, Slope of fluorescence for the third-round SEEKER with a different sample volume from the last reaction. The fluorescence enhancement in the last 120 min was considered. The error bars represent the means \pm s.d. from three replicates.

To make a more direct comparison, we calculated the fluorescence slope when the intensity started to increase across the 4 h of fluorescence monitoring. According to the real-time kinetics, for the second round of SEEKER with a matched query, the fluorescence enhancement in the first 90 min was considered. For the other experiments, the fluorescence enhancement at the last 120 min was considered. The slopes of fluorescence for the second and third round of SEEKER are displayed in **Supplementary Fig. 22f** and **Supplementary Fig. 22g**, respectively.

In this revised version, we have included the description of reusable SEEKER in Lines 323-336 of the main text and an elaborate discussion of the experimental procedures in Supplementary Fig. 22.

With regard to reproducibility, the computational results appear to be reproducible. Although, it's not written with many comments. Many of the codes provided are inefficient. A significant concern is that no listing of required software packages is provided. It's recommended that the authors provide a standard requirements.txt file as is the custom for Python code bases. Please see <https://pip.pypa.io/en/stable/reference/requirements-file-format/>.

Response: We appreciate the reviewer's response about the codes. We have modified the codes and included more comments. Moreover, we have provided a requirements.txt file in the original GitHub repository. Please see <https://github.com/Jiozhang/SEEKER-encoding-and-decoding>. We have also listed the required Python packages in README.md.

References

- 1 Bee, C. *et al.* Molecular-level similarity search brings computing to DNA data storage. *Nature Communications* 2021 12:1 **12**, 1-9 (2021). <https://doi.org:10.1038/s41467-021-24991-z>
- 2 Newman, S. *et al.* High density DNA data storage library via dehydration with digital microfluidic retrieval. *Nat Commun* **10**, 1706 (2019). <https://doi.org:10.1038/s41467-019-09517-y>
- 3 Kosuri, S. & Church, G. M. Large-scale de novo DNA synthesis: technologies and applications. *Nat Methods* **11**, 499-507 (2014). <https://doi.org:10.1038/nmeth.2918>
- 4 Erlich, Y. & Zielinski, D. DNA Fountain enables a robust and efficient storage architecture. *Science* **355**, 950-954 (2017). https://doi.org:10.1126/SCIENCE.AAJ2038/SUPPL_FILE/ERLICH.SM.PDF
- 5 Organick, L. *et al.* Random access in large-scale DNA data storage. *Nature Biotechnology* 2018 36:3 **36**, 242-248 (2018). <https://doi.org:10.1038/nbt.4079>
- 6 Lu, S. *et al.* Fast and sensitive detection of SARS-CoV-2 RNA using suboptimal protospacer adjacent motifs for Cas12a. *Nat Biomed Eng* **6**, 286-297 (2022). <https://doi.org:10.1038/s41551-022-00861-x>

Response to Reviewer 2's Comments:

We appreciate reviewer 2 for taking his/her valuable time to comment. Below, we have presented reviewer 2's comment in italics, and the corresponding responses are marked in blue font.

Reviewer #2 (Remarks to the Author):

The authors Zhang et al. investigate the viability of adopting a CRISPR-based method for keyword searches in DNA-based data storage systems. Through experiments, they find a near-linear correlation between the fluorescence intensity over a set duration and the number of queried keyword present in the file that is stored in the pool. This suggests a potential to estimate keyword counts in stored files without sequencing the entire pool. Compared to the in-tube tests, on-chip implementations produced noisier results but led to similar conclusions. Additionally, given that the method necessitates sequential text storage, the paper introduces a new coding scheme, NCG, which offers a more compact storage solution than directly storing the text information in DNA without compression. The simulations show that NCG is able to achieve a coding density slightly higher than 2 bits/nt in practical scenarios.

Response: We appreciate the reviewer's positive comments!

While the concept of "keyword-searching" in DNA-based storage is intriguing, I have the following concerns about the proposed approach.

The proposed method is still not "sequencing-free", which is supposed to be the goal of the keyword-searching problem. To construct the query, one would need to first sequence part of the pool corresponding to the reference sequences, to know the NCG encoding dictionaries. However, if sequencing is needed anyhow, why not sequence the whole pool and search in the recovered text directly? The PCR amplification is of low cost compared to DNA synthesis, and one round of sequencing should provide enough coverage for all the encoded DNA sequences, at least for the data scale shown in the paper. So what are the key benefits of using the proposed method here? On the other hand, the drawback of the proposed approach is obvious, as one cannot use the well-developed compression methods for text anymore, resulting in a significant decrease in the storage density. It would be great if the authors can explain and analyze the trade-offs of the proposed method.

Response: Thank you for this question. We agree with the reviewer's viewpoint that being "sequencing-free" is the ultimate goal in keyword searching. Due to the current cost of oligo pool synthesis, most research studies, including ours, can only propose DNA-based storage with a volume of KB to MB. In this case, a single sequencing run may be enough to cover the entire pool including the dictionary. However, with the cost of DNA synthesis and sequencing decreasing, we envision that future DNA data storage work may reach information scales of GB or even TB. In such cases, it might be unrealistic to read the complete data in one sequencing run. Additionally, since the entire oligo pool is sequenced, the decoded data may include many unwanted contents which would be both a waste of time and digital memory. In this work, we demonstrated that the dictionary under this coding scheme may occupy only 5% or less of the entire storage space in an oligo pool. Therefore, dictionary sequencing is much less of a burden than data sequencing. Combined with SEEKER, only data-of-interest, which may occupy 1% or less of the corpus, will be selected, sequenced, and decoded, greatly saving computing resources. Moreover, as fewer

DNA strands will be sequenced, the sequencing coverage can be elevated, reducing the likelihood of missing a read for a particular strand. This makes the recovery of data stored in DNA more secure and reliable.

For the second question, we understand the reviewer's concern that most coding techniques do not properly adapt to our proposed keyword search scheme. Although we agree that many conventional coding techniques may achieve a higher coding potential and better storage density, we still propose this method with the following considerations:

1. The original order of words cannot be disrupted, otherwise the keyword search would not be achievable. This excludes block-sorting compression.
2. The search is meant to be implemented prior to reading the file. Therefore, any coding techniques in which the dictionary can only be obtained after reading the complete content of the file should be excluded. This excludes some widely applied dictionary coders such as Lempel-Ziv compression and its variants.
3. It is more beneficial to use fixed-length coding if we intend to conduct keyword search molecularly. This excludes Huffman coding, arithmetic coding, and their variants. We believe fixed-length coding is a requirement not only for SEEKER but also for all molecular search approaches. SEEKER naturally limits the length of a query sequence since the spacer of crRNA may only contain 18–24 nt. However, the proper length of a query is even more important if we adopt conventional strategies such as enzyme-free DNA hybridization. This is mainly because the length of a DNA strand will greatly affect its binding affinity and specificity. If a query contains very few bases, it may not bind efficiently to its target and thus may result in a search failure. If the query sequence is too long, the efficiency and specificity of binding may become uncontrollable due to a higher risk of strong secondary structure formation in the query, the potential for homodimers to form between queries, and non-specific interactions between the query and irrelevant nucleic acids.
4. Run-length encoding, which has been widely applied to DNA data storage, may not be suitable for molecular data searches because the query sequences may be different across a single data block even if it refers to the same keyword.

As you can see, these above considerations excluded almost all the classical text data compression techniques, which we hope explains why we developed a new coding method that is compatible with SEEKER and other potential search schemes for DNA data storage.

In this revised version, we have included the above discussion and analysis in Section 2.1 of the Supplementary Information as the background for developing this coding algorithm.

Using CRISPR for text searching can be restrictive. As noted by the authors "the optimal range of length of a target being recognized by CRISPR reaction is 18-24 nt," most likely we are only able to search one word with one query. I am wondering if the method can be adapted to enable the search for phrases and even sentences. Can it be achieved by a mixture of queries treated to the pool? Also, in current experiments, the keyword frequency is a bit low (max is 5). How will the system perform when this frequency is high (i.e., 100, 1000)? Will the fitted curve shown in Figure 3 (k) still be linear?

Response: We appreciate the reviewer's questions about the potential limitations of CRISPR-based searching in terms of long queries and high keyword frequencies. Since the activation of

trans-cleavage in the CRISPR-Cas12a system only requires one perfect match of crRNA and DNA target, we considered the idea of using a mixture of queries in one CRISPR reaction. However, by setting up multiple reactions to build a search array, it is entirely possible to adapt SEEKER to search for longer keywords, phrases, and sentences. It is very likely that this kind of search array could be built on grid-patterned microchambers, with each chamber pre-storing a CRISPR reaction with an individual query as a split of the original long query. When in operation, the file amplicons may flush through the corresponding chamber row to enable mixing of the CRISPR reaction, crRNA query, and DNA target. A schematic draft of this system is shown in **Supplementary Fig. 1**.

Supplementary Fig. 1. Searching with long queries in SEEKER. By setting up multiple reactions to build a search array, it is possible to adapt SEEKER to search for longer keywords, phrases, and sentences. This search array can be created in grid-patterned microchambers pre-loaded with CRISPR reactions, with each individual short query as a split of the original long query. During operation, the file amplicons may flush through the corresponding chamber row, enabling mixing of the CRISPR reaction, crRNA query, and DNA target. The digital fluorescence state of 0/1 will indicate the presence or absence of short queries, allowing the user to infer the existence of a long query.

It should be noted that in some cases, when the paragraph or article contains the searched words but not in the order of forming a sentence, it may result in search errors. This is because the queries consist of single words and not full sentences. This problem may only be solved by a more advanced coding technique that is capable of converting variable-length codes into fixed-length DNA sequences without the need to store a huge dictionary or recover the full data before determining the dictionary. Moreover, we believe that even when using long queries in searching, it is still better to limit the length of one query oligo. This is because an ultralong single query sequence may lead to unpredictable efficiency and specificity in keyword identification.

To address the reviewer's concern about the ability to conduct quantitative searches with higher keyword frequencies, we designed an experiment using simulated keyword frequencies by adjusting the proportion of keyword-containing oligos to the entire oligo pool. This method reduces the cost of oligo pool synthesis while enabling the flexibility in obtaining different keyword frequencies. We designed two oligos that are prefixed and suffixed with the same pair of PCR primer target sequences so that they belong to one file. One of the oligos contained no keyword sequence and the other contained one keyword sequence, where the keyword sequence

corresponded to the query 'storage' as used in this study. To exclude interference generated by potentially similar sequences on the 'no keyword' oligo, we applied the CRISPR-Cas12a system to 12 nM of non-amplified oligos and observed no fluorescence response for the 'no keyword' oligo (**Supplementary Fig. 21a**).

The oligo sequences are listed in **Supplementary Table 9**, with the primer targets marked in purple and keyword sequences marked in orange:

Name	Sequence (5'-3')
crRNA-storage_1	UAAUUUCUACUCUUGUAGAUUAGCGUCACGCGUCGUCGUC
No keyword oligo	TGTCTTCTCCAGCAACGAATGGTACTGTACTGGTAGATGGCGC ACGGCGACGTGTCTATGGTGATACGCTACATGAGTCACGCTCG CAATGAGCAGTAGTCGGTCGCGCGTTCGAGTAGTGGCCAACAA TCCTACTGCCTAG
Keyword-containing oligo	TGTCTTCTCCAGCAACGAATGGTCTGACGAGCGACGCGTGACG CTAGCTCGTGATATCGCGTCGTGACAGAGTGGAGTATGATGAGC AGTATCATGACACGTGTCTGCAGATATGTGTGTAGTAGCACTAG CTATCGAGTACGCCAACAAATCCTACTGCCTAG
Forward primer	TGTCTTCTCCAGCAACGAATG
Reverse primer	CTAGGCAGTAGGATTGTTGGC

Supplementary Table 9. Oligo sequences for experiment of higher keyword frequency simulation (primer targets are marked in purple and keyword sequences are marked in orange).

As this experiment aims to simulate keyword frequencies up to 1,000, in order to simulate the condition where the keyword frequency is 1, the keyword-containing oligos must occupy 1‰ of the oligos comprising the file. Similarly, to simulate the condition where the keyword frequency is 5, the keyword-containing oligos must occupy 5‰ of the entire file. The concentration of the file should be fixed while adjusting the keyword frequencies to ensure the comparison is reasonable. In the experiment, we used 500 aM, which was equal to around 300 copies/ μ L, as the file concentration. In addition, in each reaction we introduced 50 pM of λ DNA as a noisy background representing other files in the oligo pool.

Supplementary Fig. 21. SEEKER with high keyword frequencies. **a**, Real-time fluorescence of CRISPR reactions with unamplified 12 nM of 'no keyword' and 'keyword-containing' oligos, which confirmed that 'no keyword' oligos induced no interference in fluorescence response. **b**, Real-time fluorescence with simulated keyword frequencies. A higher keyword frequency typically corresponded to a faster enhancement in fluorescence intensity, while for a frequency of 1,000 the fluorescence response was slightly weaker. **c**, The average slope of fluorescence with different keyword frequencies for the first 30 min of the reaction. The average slope increased with frequencies smaller than 500, but halted at a frequency of 1,000, potentially due to minimal differences in the generation of keyword-containing amplicons during PCR when the amount of keyword-containing oligos was high. **d**, Linear fitting between average slopes and simulated keyword frequencies below 200, with an R² value of 0.899. **e**, Linear fitting between average slopes and simulated keyword frequencies below 500, with an R² value of 0.739. These results suggest that 200 was a better keyword frequency limit for quantitative searching with SEEKER. In the experiment, to simulate the condition where the keyword frequency was 1, the keyword-containing oligos occupied 1‰ of the oligos making up the file, assuming each file contains 1,000 oligos. We used 500 aM, which was equal to around 300 copies/μL, as the file concentration. In addition, in each reaction we introduced 50 pM of λDNA as noisy background representing other files in the oligo pool. The error bars represent the means ± s.d. from four replicates. The oligo sequences used in this experiment can be found in Supplementary Table 9.

We monitored real-time fluorescence with different keyword frequencies over 120 min. We found that higher frequencies typically corresponded to a faster enhancement in fluorescence intensity, while for a frequency of 1,000 the fluorescence response was slightly weaker (**Supplementary Fig. 21b**). We then repeated the experiment three more times and calculated the average slope of fluorescence for the first 30 min of the reaction. As shown in **Supplementary Fig. 21c**, we found that the average slope increased with frequencies smaller than 500, but halted at a frequency of 1,000. This could be potentially due to minimal differences in the generation of keyword-containing amplicons during PCR when the amount of keyword-containing oligos was high. It suggests the frequency limit for keyword searching in the current SEEKER system is close to 1,000. Excluding the data where the frequency was 1,000, we performed linear fitting between the average slope and simulated keyword frequency. We found a better fitting result for frequencies below 200 (R² = 0.899) (**Supplementary Fig. 21d**) compared to frequencies below 500 (R² = 0.739) (**Supplementary Fig. 21e**). This result indicates that saturation might already occur at a frequency of 500, and that the keyword frequency limit for quantitative searching with SEEKER is 200.

*In this revised version, we have included a solution to address searches with long queries in **Lines 130-133** of the main text and in Section 1.1 of the Supplementary Information. The experiment showing quantitative searches with simulated higher keyword frequencies is mentioned in **Lines 315-318** of the main text and the detailed results can be found in **Supplementary Fig. 21**.*

Some arguments about the NCG coding are not correct. First in the SI, "Specifically, to reach Shannon capacity, we have: $n_2 \leq 4n_1 - 1$ ". If we are converting $8n_1$ bits into $n_2 - 1$ bases, the naïve relationship should be $2(n_2 - 1) \geq 8n_1$, leading to $n_2 \geq 4n_1 + 1$. And this will also affect the limit in SI equation (10), as the limit now should be smaller than 2. Furthermore, I do not think it is appropriate to analyze the asymptotic coding density based on the sigmoid assumption and empirical fitting of the parameters, and show the results of the main text. Figure 2 (a)-(f) of the main text is misleading and may not be the case in practice. If the authors want to analyze the coding density rigorously, they would need to develop probabilistic

arguments to show the lower and upper bounds on the number of groups, given assumptions on the distributions of 0 and 1 at each position in the binary sequences.

Response: We appreciate the reviewer’s comment on the method we used for the performance analysis on NCG coding. We acknowledge that it is inappropriate to estimate the number of groups by sigmoidal fitting. As the reviewer suggested, it is a good idea to estimate the upper and lower bounds of the number of groups using a probabilistic model. However, before this, we believe it is necessary to estimate the coding potential and determine the best grouping interval from a theoretical perspective based on the consideration of extreme conditions of data composition.

The foundation of NCG in data compression mainly rely on the elimination of repeated and “collided” data units by assigning them to one group, and the coding density is mainly influenced by the number of groups that further influences both the number of bases in the reference strands and data strands. Thus, the best-case scenario is when the data only contain three unique units, and these units happen to be “non-collided” and are therefore assigned to one group. The worst-case scenario is when the data contain no repeated units and no “non-collision” units are found, and thus each group only includes one data unit.

In the worst-case scenario, suppose we have N symbols, making up $8N$ bits of data in total. Given the grouping interval $G = 8n_1$, if all the data units are unique, the number of groups should be:

$$N_G = \frac{8N}{8n_1} = \frac{N}{n_1} \quad (1)$$

Then, the number of bases in the reference strands should be the same as the data size:

$$B_{rp} = 8n_1 \cdot \frac{N}{n_1} = 8N \quad (2)$$

The number of bases in the data strands will be:

$$B_{dp} = \frac{8N}{8n_1} \cdot (\log_3 N_G + 1) = \frac{N}{n_1} \left(\log_3 \frac{N}{n_1} + 1 \right) \quad (3)$$

As this analysis is for theoretical estimation of the upper and lower limit of the coding potential, we assume the number of data bases to encode one data unit changed continuously rather than by one base if the threshold is reached, and the encoding volume n_2 is no longer considered in the following analysis.

Considering each binary code has its non-collided “complement code” (i.e., “00000100” and “11111011”), the ASCII code table containing 256 symbols can be divided into two parts, the original codes and the complement codes, in which each original code can pair with its complement code and be assigned to one group, but the original codes can only be assigned to different groups. In this case, when the grouping interval is 8 ($n_1 = 1$), which means one data unit contains one ASCII code, there are $2^7 = 128$ original codes and 2^7 groups in maximum. When the data size is below or equal to 2^7 bytes, in the worst-case scenario we assume the data is all composed of original codes. Then, the coding potential when ($N \leq n_1 \cdot 2^7 = 2^7$) can be calculated as:

$$P = \frac{8N}{8N + N(\log_3 N + 1)} \quad (4)$$

When the data size is larger than 2^7 bytes, which refers to $N > 2^7$, the coding potential is:

$$P = \frac{8N}{8 \times 2^7 + N(\log_3 2^7 + 1)} \quad (5)$$

Similarly, when the grouping interval is 16 ($n_1 = 2$), there are 2^{15} original codes. When $N \leq n_1 \cdot 2^{15} = 2^{16}$, the representation of the coding potential is:

$$P = \frac{8N}{8N + \frac{N}{2} \left(\log_3 \frac{N}{2} + 1 \right)} \quad (6)$$

When $N > 2^{16}$, the coding potential is:

$$P = \frac{8N}{8 \times 2^{16} + \frac{N}{2} (\log_3 2^{15} + 1)} \quad (7)$$

We can generalize the representation of the coding potential with any n_1 as:

$$P = \begin{cases} \frac{8N}{8N + \frac{N}{n_1} \left(\log_3 \frac{N}{n_1} + 1 \right)}, & 0 < N \leq n_1 \cdot 2^{8n_1-1} \\ \frac{8N}{8n_1 \cdot 2^{8n_1-1} + \frac{N}{n_1} (\log_3 2^{8n_1-1} + 1)}, & N > n_1 \cdot 2^{8n_1-1} \end{cases} \quad (8)$$

In the best-case scenario, we assume the data are ideally composed of repeats of three “non-collided” data units, which can be exactly assigned to one group, resulting in only $8n_1$ bases in the reference strand. Then, the coding potential can be represented as:

$$P = \frac{8N}{8n_1 + \frac{2N}{n_1}} \quad (9)$$

Based on the above analysis, we can then determine the lower and upper bounds of the coding potential as a function of symbol number N and grouping interval n_1 . With data size as the X-axis, the upper and lower bounds of the coding potential under different n_1 are plotted in **Supplementary Fig. 3a**, where the value of n_1 is marked in each subfigure.

Supplementary Fig. 3. Theoretical and practical estimation of coding potential for NCG. **a**, Theoretical upper (red curves) and lower bounds (blue curves) of coding potential under different n_1 based on consideration of extreme conditions of data composition. The value of n_1 was marked in each subfigure. **b**, The lower bound of coding potential when $n_1 \leq 8$ with data size up to 1,000 MB. **c**, The actual coding potential obtained with real text data from Wikipedia when $n_1 = 1, 2, 3$. Each data size was tested using 10 different blocks as experimental replicates. **d**, The normal distribution of the number of groups when $n_1 = 2$. Block sizes were marked in each subfigure. For each block size, 100 different blocks as experimental replicates were tested. **e**, The measured results of the number of groups from the 100 replicates with different block sizes. **f**, The calculated upper and lower limit of number of groups with every block sizes based on a probabilistic estimation of normal distribution. **g**, The upper and lower limit of coding potential for every tested block size based on the estimated upper and lower limit of the number of groups.

Through comparison of the lower theoretical bounds, we were able to determine the best grouping interval and the range of data block sizes to be applied in NCG coding. With the current stage of

DNA data storage mostly involving data sizes at the MB level, we calculated the lower bound of coding potential under different n_1 with data sizes up to 1,000 MB (**Supplementary Fig. 3b**). We found in the cases of $n_1 = 1$ and $n_1 = 2$, the lower bound rapidly grew when the data size was less than 1 MB. When $n_1 = 2$, the lower bound was the highest across the studied data size range. Finally, when $n_1 = 3$ the lower bound gradually increased but was still slightly lower than the case of $n_1 = 2$ when the data size reached 1,000 MB. For higher n_1 values, the lower bounds remained significantly lower. This result suggested that $n_1 = 2$ might be a better choice as the grouping interval for NCG for text data encoding.

Then, we aimed to provide measured upper and lower bounds based on a probabilistic model using real input data, which can better reflect the actual conditions of text data encoding. To demonstrate this, we downloaded ~100 MB of text data from Wikipedia as the input. The original data were divided into multiple blocks with a certain size, and different block sizes were studied to reveal the relationship between data size and the number of groups under the NCG algorithm. For each block size, 100 blocks were computed for experimental repeats, and we expect the number of groups in these 100 experiments to follow a normal distribution. This allows us to estimate the upper and lower limits from a probabilistic perspective. The block sizes chosen were 5 KB, 10 KB, 25 KB, 50 KB, 100 KB, 250 KB, 500 KB, and 1,000 KB.

Before moving on to the measurement of the number of groups, we aimed to confirm the conclusion drawn from the theoretical analysis that $n_1 = 2$ should be applied to NCG coding. We analyzed the coding potential when the n_1 value ranged from 1 to 3 using real text data from Wikipedia, testing 10 different blocks as replicates for each size. We confirmed that $n_1 = 2$ had a better actual coding potential with data sizes below 50 KB (**Supplementary Fig. 3c**). When the data size further increased, although the coding potential for $n_1 = 3$ might increase, the number of groups would increase even more dramatically, resulting in much longer computation times and greatly reducing the efficiency. Therefore, we believe it is reasonable to use $n_1 = 2$ as the grouping interval.

With the grouping interval n_1 determined as 2, the distributions of the number of groups are shown in **Supplementary Fig. 3d**, with block sizes marked in each subfigure. For most block sizes, the number of groups in the 100 experiments followed a normal distribution. The measured results of the number of groups are also shown in **Supplementary Fig. 3e**. We observed an increase in the average number of groups as the block size increased, with what appeared to be a linear relationship with the logarithm of the block size. However, there was substantial variation in the number of groups when the block size became large, which led to even a fewer number of groups for a larger block size in some cases.

We defined the upper and lower limit as when the significance level reached 0.01, where the probability of observing a number of groups exceeding the defined upper limit or below the defined lower limit is 0.01. To calculate the upper and lower limit, we used the Z-score method:

$$Z = \frac{x - \mu}{\sigma} \quad (10)$$

where x is the observed value, μ is the sample mean and σ is the standard deviation. From the Z-score table, the Z-value should be 2.33 for a significance level of 0.01. Thus, the upper and lower limit are calculated as:

$$x_L = \mu \pm 2.33\sigma \quad (11)$$

The upper and lower limits of the number of groups with every block size tested are shown in **Supplementary Fig. 3f**. The upper and lower limits rapidly increased when the data size was smaller than 100 KB. When the data size further increased, the lower limit reached a stable value at around 1,000, while the upper limit still increased but at a slower rate. We anticipate that the upper limit will soon reach its threshold if the data size is further expanded, due to the fact that the number of frequent character combinations in the English text is limited. We also calculated the practical upper and lower limit of coding potential for every block size we tested based on the estimated upper and lower limit of the number of groups. The results are shown in **Supplementary Fig. 3g**. It should be noted that in practice, when the number of groups exceeds $3^6 = 729$, an additional base must be added to each data unit of the data strands, which may compromise the overall coding potential. Consistent with our speculation, the optimal upper limit of the coding potential was obtained with data sizes of 25–50 KB, which corresponded to a measured number of groups close to 729 (**Supplementary Fig. 3e**). As the data sizes further enlarged, the coding potential dropped back to below 2 bits/nt but kept approaching 2 bits/nt. We concluded that in this coding scheme, to achieve a higher coding potential the data size encoded with one dictionary should be limited to 25–50 KB, and more dictionaries are recommended if encoding data with larger sizes than the optimal range.

In this revised version, we have included the above analysis in Section 2.3 of the Supplementary Information.

- Besides the technical issues, the paragraphs describing the NCG coding can be improved. The authors show the performance of the coding scheme before explaining how the coding is done, which could confuse the readers. I think a more natural order should be first explaining the whole procedure clearly, then discussing the performance, otherwise, the readers will have no idea what role n_1 and n_2 are playing here and why the coding density would change with the data size.

Response: Thank you for the suggestion on the organization of the paper. We have moved a large portion of the result discussion of the NCG algorithm to the Supplementary Information, only keeping the general encoding process in the main text. In doing so, many technical terms will not be mentioned in the main text to improve the readability for a broader audience. If readers are interested in more technical elaboration and performance analysis, they will find a systematic and detailed explanation to NCG coding in the Supplementary Information.

- Some minor issues: There seem to be some typos in the manuscript. For example, in Figure 2 (i), after grouping the set of group ids is $\{640, 651, 716\}$, while for other parts of the figure it is $\{641, 650, 716\}$. Also, some flowcharts in the SI are not explained clearly. For example, in SI Figure 10, the authors want to show the procedure of finding the minimal set of non-collision groups such that the maximum group size is 3; however, the procedure shown in SI Figure 10 is misleading. I understand that the authors are trying to say they are using some greedy approach to obtain the groups, but I would recommend improving the readability of the flowcharts in the SI.

Response: We appreciate the reviewer's comments on these minor issues. We have checked for and corrected the typo errors in the manuscript. In addition, we have reorganized and simplified the flowcharts in SI, particularly the flowchart showing the greedy search of non-collision groups in a dataset. We hope the readability has improved.

Reviewers' Comments:

Reviewer #1:

Remarks to the Author:

The revised manuscript substantially addresses my concerns. My questions and comments were all adequately addressed.

I am still skeptical that the NCG algorithm is the best approach, but the additional analyses and explanations added to the supplementary material at least provide an objective evaluation and do show advantages on the cases studied. And, this makes it possible for others to better understand the work, and justifies the choices made by the authors.

I still think more could be done to explain why the NCG algorithm was designed to work the way it does in the main text. However, overall, I think the paper plus supplementary material now stands on its own and has enough analysis for future researchers to understand its strengths and limitations.

As I mentioned in my first review, I think there is a lot of novelty in figuring out how to use CRISPR for search, and the various systems described are valuable work for the community to build on. I support the publication of this manuscript.

Reviewer #2:

Remarks to the Author:

I appreciate the authors' efforts in preparing a detailed and comprehensive rebuttal. The reorganization of the main text made it clearer and easier to follow, but unfortunately, some of the new results and claims presented in the rebuttal validate our concerns. Here are some of the critical points.

Firstly, the new results showed that the relationship between RFU/min and keyword frequency is log-like instead of linear (Sup Fig 21). This set of results makes sense, but it defeats the purpose of using the proposed technique to quantitatively determine the number of keywords in the text. This should be pointed out in the main text, instead of just using the results up to word-frequency 5 in Fig 3. Furthermore, if I understand correctly, the current test (for various keyword frequencies) was performed in a synthetic setting, and the number of keywords appearing in each oligo is limited (i.e., at most 1). The case where multiple keywords appear in the same oligo should be considered and analyzed as well.

Secondly, the authors have pointed out that if we try to use multiple queries to search for a longer text, the proposed method will identify all permutations of the queried letters/words. For example, both "never give up" and "give up never" will be identified when we use queries "never" and "give up" together. This prevents us from searching for a longer text in the file and can be a serious issue for search systems. The root cause of this is mainly because the length of the query sequence is highly limited, and it seems that there is no easy way to resolve this issue, at least based on the proposed method.

Lastly, the contribution of this paper is not very clear. On one hand, this paper proposed a keyword searching mechanism implemented with CRISPR, but it requires the data to be stored in a very specific form where most compression methods are not allowed (the details are included in the author rebuttal letter p23), meaning that the proposed method is not compatible with most of the existing DNA-based storage systems; on the other, the analysis of encoding technique NCG presented in the paper is not rigorous from an information theory perspective. It is hard to have an estimate of the encoding performance, and in some cases, it seems even worse than plain vanilla encoding without any compression (Sup Fig 3). These two facts highly limit the potential of the proposed scheme and hinder the contribution of this paper.

In summary, although I think the idea of using CRISPR for keyword searching is interesting, I do not think this paper is ready for publication in Nature Communications.

Response to Reviewer #1's Comments:

We appreciate reviewer #1 for taking his/her valuable time to comment. Below, we have presented reviewer #1's comment in italics, and the corresponding responses are marked in blue font.

Reviewer #1 (Remarks to the Author):

The revised manuscript substantially addresses my concerns. My questions and comments were all adequately addressed.

I am still skeptical that the NCG algorithm is the best approach, but the additional analyses and explanations added to the supplementary material at least provide an objective evaluation and do show advantages on the cases studied. And, this makes it possible for others to better understand the work, and justifies the choices made by the authors.

I still think more could be done to explain why the NCG algorithm was designed to work the way it does in the main text. However, overall, I think the paper plus supplementary material now stands on its own and has enough analysis for future researchers to understand its strengths and limitations.

As I mentioned in my first review, I think there is a lot of novelty in figuring out how to use CRISPR for search, and the various systems described are valuable work for the community to build on. I support the publication of this manuscript.

Response: We sincerely thank the reviewer for his/her positive comments! As encouraged by the reviewer, we hope this work can be valuable for researchers exploring future techniques in the exciting field of DNA data storage, both biochemically and algorithmically, and we expect this work can lead to more perfect solutions to building a search system for data stored in DNA.

Response to Reviewer #2's Comments:

We appreciate reviewer #2 for taking his/her valuable time to comment. Below, we have presented reviewer #2's comment in italics, and the corresponding responses are marked in blue font.

Reviewer #2 (Remarks to the Author):

I appreciate the authors' efforts in preparing a detailed and comprehensive rebuttal. The reorganization of the main text made it clearer and easier to follow, but unfortunately, some of the new results and claims presented in the rebuttal validate our concerns. Here are some of the critical points.

Response: We sincerely thank the reviewer for his/her previous comments and suggestions, which has greatly improved the quality of our manuscript. We also regret that some of your concerns were not fully addressed in our last rebuttal. In this letter, we will try to make a clearer clarification on the points you raised. Please see the point-by-point response below.

Firstly, the new results showed that the relationship between RFU/min and keyword frequency is log-like instead of linear (Sup Fig 21). This set of results makes sense, but it defeats the purpose of using the proposed technique to quantitatively determine the number of keywords in the text. This should be pointed out in the main text, instead of just using the results up to word-frequency 5 in Fig 3. Furthermore, if I understand correctly, the current test (for various keyword frequencies) was performed in a synthetic setting, and the number of keywords appearing in each oligo is limited (i.e., at most 1). The case where multiple keywords appear in the same oligo should be considered and analyzed as well.

Response: Thank you for your concern on the linearity between fluorescence enhancement rate and keyword frequency. We agree with the reviewer that it is important to specify the type of quantitative relationship, whether linear or log-like, in the main text so readers can have a better understanding.

SEEKER is a keyword search method that originated from CRISPR-based nucleic acid detection. As with many other methods in the field of analytical chemistry, the linearity between signal and analyte concentration in SEEKER only applies to concentrations within the linear range¹. When analyte concentration is close to the limit of detection (LOD), the signal is supposed to be directly proportional to the analyte concentration^{2,3}. However, with a broader range of analyte concentration, the signal is more likely to be proportional to the logarithm of analyte concentration since there is always a saturation level of signal for an assay⁴⁻⁶. As mentioned in the reviewer's comments, it explains why the quantitative relationship looked more log-like, particularly when data corresponding to high keyword frequencies were added (**Revised Fig. 3l**). We then analyzed the linearity when keyword frequency was below 20, 50, 100, 200 and 500, respectively (**Revised Fig. 3m** and **Revised Supplementary Fig. 21c**), and we found the linearity became significantly worse when keyword frequency reached 500. While a perfect linear relationship was obtained when the upper limit of keyword frequency was 20, the linearity was still tolerable ($R^2 > 0.8$) for keyword frequencies up to 200. Consistent with our expectations, when the x-variable shifted to logarithm of keyword frequency, the linearity was remarkably better than that with actual keyword frequency (**Revised Fig. 3n**).

From a more practical perspective, we would like to discuss whether this linear range fulfills the requirement of searching text data with regular contents. Although there are no strict rules about keyword density, many search engine optimization (SEO) experts recommend 0.5-2.5% as the ideal keyword density in an article/webpage (<https://surfsideppc.com/keyword-density/>, <https://www.spyfu.com/blog/keyword-density/>, <https://www.infidigit.com/blog/keyword-density/>), as this ratio can help a digital search engine better understand the content and rank it accordingly. In the synthetic experiment, the payload of data strand was designed in the same way as it was in the experiment searching actual contents in **Revised Fig. 3e-k**. The average number of words encoded in each data strand in **Revised Fig. 3e-k** was 3.89, as 5,909 words were encoded in 1,519 strands in total. Since in the synthetic setting we assumed 1,000 strands were included in the file, the number of words this file contained were estimated 3,890. Given the recommended keyword density of 0.5-2.5%, the keyword frequency in this synthetic file should be between 19.45 to 97.25. Herein, we roughly estimate that optimum range of keyword frequency in this synthetic setting is 20-100. Based on the analysis in the previous paragraph, we believe the linearity between RFU/min and keyword frequency under this range was still acceptable.

We also agree with the reviewer that investigation on the cases where multiple keywords appear in the same oligo is needed, as a recent work reported that using crRNAs targeting multiple sequences along a single DNA strand may result in enhancement of the signal^{7,8}, which suggests that the performance may be different if the oligo being searched contains multiple identical keywords rather than just one, even if the overall keyword frequency remains the same. We therefore designed two new keyword-containing oligos (see **Revised Supplementary Table 9** for sequences) including a strand containing 2 keywords distributed at two sides of payload and a strand containing 5 keywords covering the entire payload. We first tested the performance of SEEKER under two conditions: 1) when the synthetic file only contained oligos with 1 keyword and 2) when the synthetic file only contained oligos with 2 or 5 keywords. The simulated keyword frequency was kept the same (either 2 or 5 in total) in both conditions. We observed that while the fluorescence kinetics for the synthetic file composed of merely 2-keyword oligos was not apparently different from that with 1-keyword oligos (**Revised Supplementary Fig. 21d**), there was a huge variation when 5-keyword oligos were applied (**Revised Supplementary Fig. 21e**), which was consistent with the conclusion from previous reports^{7,8} that more targeting sites along the DNA strand will boost the signal.

However, in practical scenarios there is little chance that one keyword appears consecutively in a sentence. From the 0.5-2.5% keyword density, on average the keyword may appear once every 40 to 200 words, far exceeding the number of words one oligo can encode in our coding scheme. Despite this, we cannot exclude some cases where the keyword is frequently repeated in one sentence for certain purposes, especially when the keyword density in the text is high. Therefore, we conducted a new experiment investigating the limit of keyword frequency that SEEKER can identify using a mixture of no keyword, 1-keyword, 2-keyword and 5-keyword oligos simulating different keyword frequencies. The makeups of every keyword frequency are listed in Supplementary Table 10. The results in **Revised Fig. 3o** showed that although 5-keyword oligos were introduced in the group with a keyword frequency equal to or greater than 100, the signal was not dramatically increased as when pure 5-keyword oligos were tested in **Revised Supplementary Fig. 21e**, possibly due to large proportion of enzymes consumed by massive 1-keyword and 2-keyword oligos, resulting in approximate fluorescence enhancement rates for every simulated keyword frequency compared with that in experiments with merely 1-keyword oligos. Meanwhile, consistent with the results from 1-keyword experiments, an acceptable

linearity ($R^2 > 0.8$) between RFU/min and keyword frequency could still be obtained when keyword frequency was below 200 (**Revised Fig. 3p, Revised Supplementary Fig. 21g**). As the frequency became higher, the response curve looked more log-like and the linearity was good when the x-variable was the logarithm of keyword frequency (**Revised Fig. 3q**). These results indicate that in DNA data storage, a sparse amount of oligos containing many keywords will not significantly influence the ability of SEEKER in quantitatively estimating the keyword frequency.

Revised Fig. 3: Performance of SEEKER in single-file and multi-file oligo pools. a, Real-time

fluorescence kinetics of a keyword search in unamplified oligo pools at a final concentration of 600 pM. Four queries ('comput', 'CRISPR', 'COVID', and 'storage') were used to search File A and File B. **b**, Average slopes of fluorescence enhancement at 120 min for all searches in **a**. Readouts under blue LED are shown in corresponding side images. Red numbers refer to the frequency of a keyword. **c-d**, Real-time fluorescence kinetics of a search in an oligo pool referring to File B by the query 'COVID' without amplification (**c**) and with Strand #2, which contained the keyword 'COVID' amplified (**d**). Template concentrations are specified in the legend. **e-f**, Real-time fluorescence kinetics of searching for the keywords 'storage' (**e**) and 'database' (**f**) in amplified File #0. The oligo pool was serially diluted before performing PCR to determine the minimum template concentration required for SEEKER. **g**, Real-time fluorescence kinetics of searching in amplified Files #0–#7 using four keywords ('storage', 'database', 'crypto', and 'comput'). **h**, Error rates in target file identification across different reaction times. **i**, Average slopes of fluorescence enhancement for all searches in 40 files over 20 min. Red boxes indicate misidentified files under specific queries. **j**, Frequency of each keyword in every encoded file. **k**, Linear correlation between the average slope of fluorescence enhancement and the frequency of the keyword. **l**, The average slope of fluorescence with different keyword frequencies simulated with 1-keyword oligos. **m**, Linear fitting between average slopes and simulated keyword frequencies using 1-keyword oligos when keyword frequency was below 20. **n**, Linear fitting between average slopes and logarithms of simulated keyword frequencies with base 10 using 1-keyword oligos when keyword frequency was below 500. **o**, The average slope of fluorescence with different keyword frequencies simulated with a mixture of 1-keyword, 2-keyword and 5-keyword oligos. **p**, Linear fitting between average slopes and simulated keyword frequencies using multi-keyword oligos when keyword frequency was below 20. **q**, Linear fitting between average slopes and logarithms of simulated keyword frequencies with base 10 using multi-keyword oligos when keyword frequency was below 500. For **l-q**, the slopes represent the average fluorescence enhancement in the first 30 min of reaction.

Revised Supplementary Fig. 21. SEEKER with high keyword frequencies. **a**, Real-time fluorescence of CRISPR reactions with unamplified 12 nM of no keyword and 1-keyword oligos, which confirmed that no keyword oligos induced no interference in fluorescence response. **b**, Real-time fluorescence with keyword frequencies simulated with 1-keyword oligos. A higher keyword frequency typically corresponded to a faster enhancement in fluorescence intensity, while for a frequency of 1,000 the fluorescence response was

slightly weaker. **c**, Linear fitting between average slopes and simulated keyword frequencies using 1-keyword oligos when keyword frequency was below 50, 100, 200 and 500. The result suggests that a maximum keyword frequency of 200 may still lead to a tolerable linear relationship between fluorescence enhancement rate and keyword frequency, but with higher keyword frequencies the linearity might be significantly impaired. **d**, The comparison of fluorescence responses between amplified files containing merely 1-keyword oligos and merely 2-keyword oligos. There was no apparent variation in fluorescence kinetics between these two conditions. **e**, The comparison of fluorescence responses between amplified files containing merely 1-keyword oligos and merely 5-keyword oligos. The fluorescence response for file composed of merely 5-keyword oligos was significantly stronger than that for file composed of merely 1-keyword oligos. **f**, Real-time fluorescence with keyword frequencies simulated with a mixture of 1-keyword, 2-keyword and 5-keyword oligos. Similar to the results with keyword frequencies simulated with 1-keyword oligos, as the keyword frequency increased, the fluorescence response became stronger accordingly. When keyword frequency reached 1,000, the fluorescence response slightly weakened. **g**, Linear fitting between average slopes and simulated keyword frequencies using multi-keyword oligos when keyword frequency was below 50, 100, 200 and 500. Similar to the condition when merely using 1-keyword oligos, a relatively better linear relationship between fluorescence enhancement rate and keyword frequency was obtained when keyword frequency was not exceeding 200.

Name	Sequence (5'-3')
crRNA-storage_1	UAAUUUCUACUCUUGUAGAUUAGCGUCACGCGUCGUCGUC
No keyword oligo	TGTCTTCTCCAGCAACGAATGGTACTGTACTGGTAGATGGCGC ACGGCGACGTGTCTATGGTGATACGCTACATGAGTCACGCTCG CAATGAGCAGTAGTCGGTCGCGCGTCGCAGTAGTGGCCAACAA TCCTACTGCCTAG
1-keyword oligo	TGTCTTCTCCAGCAACGAATGGTCTGACGAGCGACGCGTGACG CTAGCTCGTGATATCGCGTCGTCAGACAGTGGAGTATGATGAGC AGTATCATGACACGTGTCTGCAGATATGTGTGTAGTAGCACTAG CTATCGAGTACGCCAACAAATCCTACTGCCTAG
2-keyword oligo	TGTCTTCTCCAGCAACGAATGGTCTGACGAGCGACGCGTGACG CTAGCTCGTGATATCGCGTCGTCAGACAGTGGAGTATGATGAGC AGTATCATGACACGTGTCTGCAGACGAGCGACGCGTGACGCTA GCTATCGAGTACGCCAACAAATCCTACTGCCTAG
5-keyword oligo	TGTCTTCTCCAGCAACGAATGGTCTGACGAGCGACGCGTGACG CTAGACGAGCGACGCGTGACGCTAGACGAGCGACGCGTGACG CTAGACGAGCGACGCGTGACGCTAGACGAGCGACGCGTGACG CTAGCTATCGAGTACGCCAACAAATCCTACTGCCTAG
Forward primer	TGTCTTCTCCAGCAACGAATG
Reverse primer	CTAGGCAGTAGGATTGTTGGC

Revised Supplementary Table 9. Oligo sequences for the experiment investigating the linear range of SEEKER (primer targets are marked in purple and keyword sequences are marked in orange).

Total keyword frequency	Keyword frequency contributed by		
	1-keyword oligos	2-keyword oligos	5-keyword oligos
1	1	0	0
2	2	0	0
5	5	0	0
10	8	2	0

20	16	4	0
50	40	10	0
100	75	20	5
200	150	40	10
500	375	100	25
1000	750	200	50

Supplementary Table 10. The breakup of each simulated keyword frequency used in the experiment investigating the linear range of SEEKER by a mixture of 1-keyword, 2-keyword and 5-keyword oligos.

In this revised version, we have included the results of experiments investigating the performance of SEEKER with higher keyword frequencies using 1-keyword oligos in Revised Fig. 3l-n, as well as the results obtained with mixed 1-keyword, 2-keyword and 5-keyword oligos in Revised Fig. 3o-q and Revised Supplementary Fig. 21, with explanatory sentences added in Lines 319-333 of the main text. The oligo sequence used in this experiment can be found in Revised Supplementary Table 9 and the oligo composition to make up each simulated keyword frequency can be found in Supplementary Table 10.

Secondly, the authors have pointed out that if we try to use multiple queries to search for a longer text, the proposed method will identify all permutations of the queried letters/words. For example, both "never give up" and "give up never" will be identified when we use queries "never" and "give up" together. This prevents us from searching for a longer text in the file and can be a serious issue for search systems. The root cause of this is mainly because the length of the query sequence is highly limited, and it seems that there is no easy way to resolve this issue, at least based on the proposed method.

Response: We thank the reviewer for raising this concern on distinguishing phrases with word permutations. This is a crucial consideration in developing a keyword search system especially with the feature of searching long text. In the previous response, we brought up one possible solution which was to adopt alternative coding techniques capable of converting variable-length codes into fixed length DNA sequences. However, we apologize that at that time we did not realize there was actually an easier way to address this problem, which is simply through detecting the junctions of words. To be specific, to distinguish the example phrases "never give up" and "give up never", we can use the query "er_giv" (or "ver_gi") to confirm the presence of junction between words "never" and "give", and use the query "up_nev" (or "p_neve") to confirm the presence of junction between words "up" and "never" ("_" refers to blank space in phrases) (**Revised Supplementary Fig. 1**). The coding scheme used in SEEKER supports this way of searching as we encoded every two text symbols into a 7-nt sequence segment, instead of encoding each word into a fixed-length sequence segment. We also realized that we may be able to search more flexibly by avoiding encoding the complete word but short consecutive symbols.

Revised Supplementary Fig. 1. Searching with long queries in SEEKER. **a**, By setting up multiple reactions to build a search array, it is possible to adapt SEEKER to search for longer keywords, phrases, and sentences. This search array can be created in grid-patterned microchambers pre-loaded with CRISPR reactions, with each individual short query as a split of the original long query. During operation, the file amplicons may flush through the corresponding chamber row, enabling mixing of the CRISPR reaction, crRNA query, and DNA target. The digital fluorescence state of 0/1 will indicate the presence or absence of short queries, allowing the user to infer the existence of a long query. **b**, The distinguishment of long queries with permutations of words can be achieved by detecting the “junctions” of words where word permutations differ from each other. For instance, by searching with the query “er_giv” (or “ver_gi”) we can identify the existence of “never give up”, while by searching with the query “up_nev” (or p_neve”) we can identify “give up never”.

*We apologize again for not making a clear clarification in the last rebuttal regarding this concern. In the revised version, we have modified the **Supplementary Fig. 1** to include **Supplementary Fig. 1b**. We have also removed the previous claim about the unsuitability of SEEKER in searching phrases that may contain word permutations and mentioned the way to distinguish word permutations by detecting the “word junctions” in **Lines 133-135** of the main text and **Section 1.1** of the Supplementary Information (**Page 2**).*

Lastly, the contribution of this paper is not very clear. On one hand, this paper proposed a keyword searching mechanism implemented with CRISPR, but it requires the data to be stored in a very specific form where most compression methods are not allowed (the details are included in the author rebuttal letter p23), meaning that the proposed method is not compatible with most of the existing DNA-based storage systems; on the other, the analysis of encoding technique NCG presented in the paper is not rigorous from an information theory perspective. It is hard to have an estimate of the encoding performance, and in some cases, it seems even worse than plain vanilla encoding without any compression (Sup Fig 3). These two facts highly limit the potential of the proposed scheme and hinder the contribution of this paper.

In summary, although I think the idea of using CRISPR for keyword searching is interesting, I do not think this paper is ready for publication in Nature Communications.

Response: Thank you for your concern regarding the contribution of this work. In the last revised version of the manuscript, we have transferred a large portion of the evaluation of NCG algorithm from main text to Supplementary Information since we realized the main contribution of this work is not on an algorithmic aspect. As a team focusing on developing molecular methods for interesting applications involving synthetic DNA, we aim to establish experimental protocols to address the keyword search problem in DNA data storage, which is an important topic to explore before DNA data storage becomes an accessible technique. The CRISPR-based *trans*-cleavage mechanism has gained much attention in pathogenic DNA/RNA detection owing to 1) the rapidity of generating visible results, 2) the high sensitivity of detecting nucleic acids down to aM level, 3) the mild working condition and the feasibility of being integrated in microfluidics for massive parallel detection and 4) the wide commercialization of all components of the reaction (e.g., enzyme, buffer, customized crRNA, fluorescently-labeled DNA reporter, etc.). We think these features are also the key considerations of building a useful search system for DNA data storage, as fundamentally there are few differences between searching a DNA-encoded keyword in a synthetic oligo pool and “searching” a pathogenic sequence in nucleic acid extractions of body fluids, which may contain an even more complex DNA background. From the above, we hope the reviewer can understand that the major contribution of this work is to introduce the CRISPR-based molecular search system into DNA data storage, which provides a rapid and reliable way for users with no prior knowledge of the content encoded in an oligo library to identify the file-of-interest.

The reviewer further pointed out that in some cases the NCG algorithm lost the compression capability, as the coding density was not better than the plain vanilla coding. Here we suppose the plain vanilla method the reviewer mentioned is a simple mapping of binary codes to nucleobases (e.g., “00” to “A”, “01” to “G”, “10” to “C” and “11” to “T”). In this way, the theoretical coding potential can reach 2 bits/nt since two bits are mapped to one nucleotide. We understand that in many cases shown in Supplementary Fig. 3, the coding potentials were lower than 2 bits/nt and in this way the data seemed not compressed. However, this kind of coding method is typically not accepted as it easily results in homopolymers (>3) and unbalanced GC content which are undesirable in DNA synthesizing⁹ and sequencing¹⁰. An alternative method which has been widely adopted in DNA data storage is through base-3 conversion of the original binary data followed by homopolymer-free mapping¹¹ of ternary codes to bases, which was the same technique applied in the mapping of group index to bases in reference and data strands in our work. Since this method encodes data in ternary form instead of quaternary form (i.e., DNA with four distinct bases), the theoretical coding potential of this method is $\log_2 3 \approx 1.58$ bits/nt. In this case, the estimated lower bounds of coding potential of NCG method were always higher than that of the plain vanilla

method, regardless of the input data size (**Supplementary Fig. 3g**). Despite that, we acknowledged that the coding potential of NCG algorithm is heavily influenced by the input data size and an encoding volume ($V = 3^{n_2}$, where n_2 is the number of bases in one data unit) which defines the upper limit of number of groups that can be included in one reference set. Therefore, in practical scenarios, we still encourage the users to manage the number of groups included in one reference set as close as possible to the defined encoding volume to achieve the maximum coding potential (**Page 12 of the Supplementary Information**). In addition, as shown in **Supplementary Fig. 3g**, the upper limit of coding potential exceeded 2 bits/nt when the input size was between 25–50 KB, meaning the data can be compressed even compared with the simple mapping of two binary digits to one nucleobase not considering avoidance of homopolymers. Therefore, we suggest that in this coding scheme, the data size encoded with one dictionary should ideally be limited to 25–50 KB, and more dictionaries are recommended if encoding data with larger sizes than the optimal range.

Supplementary Fig. 3. Theoretical and practical estimation of the coding potential for NCG. g, The upper and lower limit of coding potential for every tested block size based on the estimated upper and lower limit of the number of groups when $n_1 = 2$.

*In this revised version, we have discussed the improved coding potential over the homopolymer-free mapping method as well as the simple two-digits to one-base mapping method in **Section 2.3 of the Supplementary Information (Page 12)**. We also mentioned in **Lines 277-279 of the main text** that the number of groups should be close to the defined encoding volume to achieve a higher coding potential.*

Furthermore, we would like to discuss some more advantages and potentials SEEKER can bring to the field of DNA data storage.

1) Enhanced specificity in keyword identification empowered by CRISPR

A reliable keyword search system typically requires the keyword to be accurately identified. In a DNA-encoded database, this requirement implies that an interfering sequence with only a few mismatches to the target sequence should not be identified. However, existing DNA storage systems relied heavily on spontaneous hybridization-based approaches to demonstrate searching capabilities^{12,13}, which may cause severe misidentifications since non-specific hybridization may occur when two sequences share much similarity. We speculate that the SEEKER method utilizing the specificity given by CRISPR-Cas12a may exhibit an enhanced identification accuracy.

To prove this, we designed experiments to compare the performance of the conventional DNA hybridization method and CRISPR-based method in recognizing mismatched sequences. The query is a 20-nt ssDNA corresponding to the keyword 'storage' as used in our study. In a simplified experimental design, we used this query to recognize different targets, including the fully complementary one and targets with continuous four-base mismatches. To avoid massive continuous matches between the query and a mismatched target, which could more possibly generate non-specific signals for both the conventional DNA hybridization and the SEEKER method, the mismatched bases were designed in the middle of the target sequence. The sequences of the query and target DNA, either matched or mismatched, are listed in **Supplementary Table 8** (red domains indicate the mismatched bases):

Name	Sequence (5'-3')
crRNA-storage_1	UAAUUUCUACUCUUGUAGAUUAGCGUCACGCGUCGCUCGUC
Fully matched DNA target	GACGAGCGACGCGTGACGCTA
Mutation_1	GACGACGCTCGCGTGACGCTA
Mutation_2	GACGAGCGAGCGCTGACGCTA
Mutation_3	GACGAGCGACGCGACTGCTA

Supplementary Table 8. Oligo sequences for experiment comparing the specificity between SEEKER and the conventional hybridization approach (red bases refer to mismatches).

For the SEEKER method, the specificity was measured through real-time monitoring of fluorescence generated from CRISPR-based trans-cleavage of ssDNA reporters, the same mechanism as introduced in this study for keyword identification. For the conventional hybridization method, we mixed the query with the targets, incubated the reaction at room temperature for 1 h, and performed native gel electrophoresis to measure the extent of binding between the query and targets. In both approaches, the concentration of the query and all target species was fixed to 500 nM with a 1:1 ratio.

Supplementary Fig. 19. Comparison of specificity in sequence identification between SEEKER and the hybridization-based approach. **a**, The normalized trans-cleavage rate for mismatched and fully matched targets with SEEKER. The average trans-cleavage rate of three experimental repeats for the fully matched target was normalized to 1.0. The results indicate that all mismatched targets had a relative trans-cleavage rate significantly lower than the fully matched one, where the highest among them reached a value of 0.2. **b**, Gel electrophoresis showing the binding affinity between the query and targets. The band indicating hybridization between mutated target-2 and the query was even darker than that in the reaction between the fully matched target and the query. **c**, Binding affinity revealed by measuring the band intensity. The band intensity for mutation-2 was almost four times the intensity for the fully matched target, suggesting a possibly much stronger binding affinity between the query and a mismatched target. The error bars

represent the means \pm standard deviation (s.d.) from three experimental replicates or measurements of the band intensity. The list of oligo sequences used in this experiment can be found in Supplementary Table 8.

For the SEEKER method, we calculated the trans-cleavage rate for every target, which was defined as the average fluorescence enhancement per minute in the initial stage of the reaction. We normalized the trans-cleavage rates in which the trans-cleavage rate for the fully matched target was the maximum (1.0). The results are shown in **Supplementary Fig. 19a**. From the results, it is clear that all mismatched targets had a significantly lower trans-cleavage rate than the fully matched one, with the highest among them reaching a value of 0.2. For the conventional hybridization method, the gel image is shown in **Supplementary Fig. 19b**, and we observed that the band indicating hybridization between mutated target-2 and the query was even darker than that in the reaction between the fully matched target and the query. To better characterize this, we measured the intensity of the hybridization bands using ImageJ and found that the band intensity for mutation-2 was almost four times the intensity for the fully matched target (**Supplementary Fig. 19c**), suggesting a possibly much stronger interaction between the query and a mismatched target.

Apparently, catastrophic misidentification in keywords or features is more likely to happen when applying the conventional hybridization method. This problem will become especially serious if the system is designed to have low tolerance in recognizing extremely similar sequences but should only recognize the exact one, given two words common in some letters are encoded in sequences with overlaps. For instance, ‘father’ and ‘mother’ share ‘ther’ at the end. However, enzyme-based methods like SEEKER will alleviate this problem by improving the specificity in sequence recognition. From this perspective, SEEKER may also make sequence design less challenging, as the requirement to find “extremely” different sequences becomes less stringent, allowing easier, more logical, and possibly more computable mapping of features to sequences.

*In this revised version, we have included these results in **Supplementary Fig. 19** and mentioned the superiority of SEEKER over hybridization-based search approaches in terms of recognition specificity in **Lines 307-310** of the main text.*

2) Apply algorithms compatible with CRISPR-based searching to metadata encoding while maintaining existing DNA storage systems for actual content encoding

The reviewer pointed out that one shortcoming of CRISPR-based searching is the incompatibility with many existing encoding methods which have been rigorously proven effective in data compression. In this study, we encoded all the text data with a CRISPR-compatible algorithm (i.e., NCG) and demonstrated CRISPR-based searching over the complete content stored in DNA to show the potential of CRISPR in searching with a complex background. Given the 40 abstracts we encoded contained an average of ~ 200 words per abstract, we estimate that there were $\sim 8,000$ interfering keywords comprising the background (**Lines 33-35 in the main text**), which was an amount never demonstrated in previous studies on search systems for DNA data storage. Considering the specificity of CRISPR-based searching in recognizing at least four base mismatches in a 21-nt target (**Supplementary Fig. 19**), a rough estimation is that SEEKER can theoretically distinguish one feature from a background of $4^{21} - C(21,3) \times 4^3 \approx 4.4 \times 10^{12}$ distinct features. However, in digital data storage systems, search is often performed in metadata which includes descriptive information about the actual data to avoid accessing enormous amounts of the actual data file-by-file. Since the size of metadata is supposed to be significantly smaller than

the size of complete data, we believe it is reasonable that in the future design, metadata is extracted from the actual data and encoded in a CRISPR-compatible algorithm to facilitate CRISPR-based searching, with little compromise in the coding density since the data size is comparatively small, while the actual data is encoded with a more widely recognized algorithm, perhaps the DNA fountain codes¹⁴ which to our knowledge had the best coding potential.

Herein, we provide a modified scheme of SEEKER that can be adapted to metadata search (**Supplementary Fig. 2a**), where the locations of a file that may carry the keyword was encoded in oligos employing the inverted indexing data structure.

Supplementary Fig. 2. Envisioning SEEKER in more advanced search settings. a, In a metadata search scheme, the features of a file, as the metadata, are pre-computed and encoded in a DNA strand along with the file index to assemble the payload. Each oligo is prefixed and suffixed with a set of PCR primer targeting sites to enable enrichment of the file. SEEKER can still be applied in this condition. When the query crRNA recognizes a matching feature, there will be fluorescence response in this enriched oligo pool, and no fluorescence will be observed if no matching feature is found. For the enriched pool with fluorescence signals, we can sequence the original pool using a small volume, or we can directly sequence the CRISPR reaction after SEEKER.

In this scheme, the features of a file, as the metadata, are pre-computed and encoded in a DNA strand along with the file index to assemble the payload. Each oligo is prefixed and suffixed with a set of PCR primer targeting sites to enable enrichment of the file. An advantage of SEEKER under this search scheme is its ability to rapidly determine the physical location of the file containing the feature. When the query crRNA recognizes a matching feature, there will be prompt fluorescence response in this enriched oligo pool, and no fluorescence will be observed if no matching feature is found. For the enriched pool with fluorescence signals, we can then perform sequencing to read the metadata. The decoding of sequencing reads will reveal which file contains the feature, and then we can read these particular files encoded with some other algorithms from the database.

*In this revised version, we have included the proposed metadata search scheme in **Supplementary Fig. 2a** and **Lines 146-149** of the main text.*

We hope by highlighting the above perspectives in the manuscript, the readers will gain a more complete understanding and find more value of the SEEKER method. We will be grateful for your consideration of the significance of this work as the first study exploring the potential of CRISPR-based searching in DNA data storage. Though there are still some imperfections in terms of algorithmic and experimental design, we expect this work can be valuable for future researchers to improve upon.

References

- 1 Araujo, P. Key aspects of analytical method validation and linearity evaluation. *Journal of Chromatography B* **877**, 2224-2234 (2009). [https://doi.org:https://doi.org/10.1016/j.jchromb.2008.09.030](https://doi.org/10.1016/j.jchromb.2008.09.030)
- 2 Zhang, D. *et al.* CRISPR/Cas12a-Mediated Interfacial Cleaving of Hairpin DNA Reporter for Electrochemical Nucleic Acid Sensing. *ACS Sensors* **5**, 557-562 (2020). [https://doi.org:10.1021/acssensors.9b02461](https://doi.org/10.1021/acssensors.9b02461)
- 3 Smith, C. W., Kachwala, M. J., Nandu, N. & Yigit, M. V. Recognition of DNA Target Formulations by CRISPR-Cas12a Using a dsDNA Reporter. *ACS Synthetic Biology* **10**, 1785-1791 (2021). [https://doi.org:10.1021/acssynbio.1c00204](https://doi.org/10.1021/acssynbio.1c00204)
- 4 Niu, C., Wang, C., Li, F., Zheng, X., Xing, X. & Zhang, C. Aptamer assisted CRISPR-Cas12a strategy for small molecule diagnostics. *Biosensors and Bioelectronics* **183**, 113196 (2021). [https://doi.org:https://doi.org/10.1016/j.bios.2021.113196](https://doi.org/10.1016/j.bios.2021.113196)
- 5 Ma, L., Peng, L., Yin, L., Liu, G. & Man, S. CRISPR-Cas12a-Powered Dual-Mode Biosensor for Ultrasensitive and Cross-validating Detection of Pathogenic Bacteria. *ACS Sensors* **6**, 2920-2927 (2021). [https://doi.org:10.1021/acssensors.1c00686](https://doi.org/10.1021/acssensors.1c00686)
- 6 Cao, G. *et al.* Completely Free from PAM Limitations: Asymmetric RPA with CRISPR/Cas12a for Nucleic Acid Assays. *ACS Sensors* **8**, 4655-4663 (2023). [https://doi.org:10.1021/acssensors.3c01686](https://doi.org/10.1021/acssensors.3c01686)
- 7 Zeng, M. *et al.* Harnessing Multiplex crRNA in the CRISPR/Cas12a System Enables an Amplification-Free DNA Diagnostic Platform for ASFV Detection. *Analytical Chemistry* **94**, 10805-10812 (2022). [https://doi.org:10.1021/acs.analchem.2c01588](https://doi.org/10.1021/acs.analchem.2c01588)
- 8 Chen, Y. *et al.* A CRISPR-Cas12a-based assay for one-step preamplification-free detection of viral DNA. *Sensors and Actuators B: Chemical* **399**, 134813 (2024). [https://doi.org:https://doi.org/10.1016/j.snb.2023.134813](https://doi.org/10.1016/j.snb.2023.134813)
- 9 Schwartz, J. J., Lee, C. & Shendure, J. Accurate gene synthesis with tag-directed retrieval of sequence-verified DNA molecules. *Nature Methods* **9**, 913-915 (2012). [https://doi.org:10.1038/nmeth.2137](https://doi.org/10.1038/nmeth.2137)
- 10 Ross, M. G. *et al.* Characterizing and measuring bias in sequence data. *Genome Biology* **14**, R51 (2013). [https://doi.org:10.1186/gb-2013-14-5-r51](https://doi.org/10.1186/gb-2013-14-5-r51)
- 11 Goldman, N. *et al.* Towards practical, high-capacity, low-maintenance information storage in synthesized DNA. *Nature* **494**, 77-80 (2013). [https://doi.org:10.1038/nature11875](https://doi.org/10.1038/nature11875)
- 12 Bee, C. *et al.* Molecular-level similarity search brings computing to DNA data storage. *Nature Communications* **12**, 1-9 (2021). [https://doi.org:10.1038/s41467-021-24991-z](https://doi.org/10.1038/s41467-021-24991-z)
- 13 Banal, J. L. *et al.* Random access DNA memory using Boolean search in an archival file storage system. *Nature Materials* **20**, 1272-1280 (2021). [https://doi.org:10.1038/s41563-021-01021-3](https://doi.org/10.1038/s41563-021-01021-3)
- 14 Erlich, Y. & Zielinski, D. DNA Fountain enables a robust and efficient storage architecture. *Science* **355**, 950-954 (2017). [https://doi.org:10.1126/SCIENCE.AAJ2038/SUPPL_FILE/ERLICH.SM.PDF](https://doi.org/10.1126/SCIENCE.AAJ2038/SUPPL_FILE/ERLICH.SM.PDF)

Reviewers' Comments:

Reviewer #1:

Remarks to the Author:

Reviewer 2 raised several concerns with the paper in their previous response.

1) Paraphrasing Reviewer 2, they wrote, "New results showed a relationship between RFU/min and keyword frequency that is log-like instead of linear. This makes it hard to quantitatively determine the number of matches. This should be pointed out. Also, the case where multiple keywords appear in the same oligo should be considered."

The authors modified the text to make this relationship clear and they added new experiments evaluating the case of the same keyword appearing multiple times within an oligo. The authors also provide substantial analysis in the rebuttal about the likelihood of such keywords appearing repeatedly in text. It's clear the authors recognize the limits of their approach but also provide a good argument for the approach taken. In my opinion, this criticism has been adequately rebutted by the authors.

2) Reviewer 2 pointed out that using multiple queries will result in finding all combinations of keywords and not a specific ordering.

The authors respond to this by offering a new kind of search query that identifies the junction of two words so that their ordering can be identified. This is a reasonable strategy, however, this raises concerns about the probability of matching the junction when the desired words are not present because it happens to match common prefixes and suffixes. A more robust analysis is needed to understand the limits of this approach. However, I would not insist on a new analysis; rather, I would ask the reviewers to acknowledge the limits of this approach and that more analysis is needed to understand how well it works in a scaled-up system.

3) Reviewer 2 has concerns about the contribution and how the work fits. Most of the concerns are directed at the encoding. It's pointed out that the encoding is not rigorous from an information theory perspective.

The authors respond that the strength of the work is in the molecular methods, and the bulk of the discussion of the encoding algorithm is moved to supplementary material. This reviewer agrees with the authors and doesn't think the relatively weaker treatment of encoding should prevent publication.

The authors have also gone to some lengths to introduce new results into the paper demonstrating a meta-data search scheme, which adds to their contribution.

Overall, I'm satisfied with the authors' response to the criticisms raised by Reviewer 2. I would recommend publication.

Response to Reviewer 1's Comments:

We appreciate Reviewer 1 for taking their valuable time to comment. Below, we have presented Reviewer 1's comments in italics and marked the corresponding responses in blue font.

Reviewer #1 (Remarks to the Author):

Reviewer 2 raised several concerns with the paper in their previous response.

1) Paraphrasing Reviewer 2, they wrote, "New results showed a relationship between RFU/min and keyword frequency that is log-like instead of linear. This makes it hard to quantitatively determine the number of matches. This should be pointed out. Also, the case where multiple keywords appear in the same oligo should be considered."

The authors modified the text to make this relationship clear and they added new experiments evaluating the case of the same keyword appearing multiple times within an oligo. The authors also provide substantial analysis in the rebuttal about the likelihood of such keywords appearing repeatedly in text. It's clear the authors recognize the limits of their approach but also provide a good argument for the approach taken. In my opinion, this criticism has been adequately rebutted by the authors.

Response: Thank you for your positive comments.

2) Reviewer 2 pointed out that using multiple queries will result in finding all combinations of keywords and not a specific ordering.

The authors respond to this by offering a new kind of search query that identifies the junction of two words so that their ordering can be identified. This is a reasonable strategy, however, this raises concerns about the probability of matching the junction when the desired words are not present because it happens to match common prefixes and suffixes. A more robust analysis is needed to understand the limits of this approach. However, I would not insist on a new analysis; rather, I would ask the reviewers to acknowledge the limits of this approach and that more analysis is needed to understand how well it works in a scaled-up system.

Response: Thank you for your comments and suggestion. We included some discussion in the revised manuscript.

3) Reviewer 2 has concerns about the contribution and how the work fits. Most of the concerns are directed at the encoding. Its pointed out that the encoding is not rigorous from an information theory perspective.

The authors respond that the strength of the work is in the molecular methods, and the bulk of the discussion of the encoding algorithm is moved to supplementary material. This reviewer agrees with the authors and doesn't think the relatively weaker treatment of encoding should prevent publication.

The authors have also gone to some lengths to introduce new results into the paper demonstrating a meta-data search scheme, which adds to their contribution.

Overall, I'm satisfied with the authors' response to the criticisms raised by Reviewer 2. I would recommend publication.

Response: Thank you for your positive comments.